# Versatile Transferable Unlearnable Example Generator

**Zhihao Li**[1,*]    **Jiale Cai**[1,*]    **Gezheng Xu**[1,†]    **Hao Zheng**[1,2]    **Qiuyue Li**[3]
**Fan Zhou**[3]    **Shichun Yang**[3]    **Charles Ling**[1,4]    **Boyu Wang**[1,4,†]

[1] Western University    [2] Central South University    [3] Beihang University    [4] Vector Institute
{zli3446,jcai336,gxu86,charles.ling}@uwo.ca   zhenghaocsu@gmail.com
{liqiuyue23,zhoufan,yangshichun}@buaa.edu.cn   bwang@csd.uwo.ca

## Abstract

The rapid growth of publicly available data has fueled deep learning advancements but also raises concerns about unauthorized data usage. Unlearnable Examples (UEs) have emerged as a data protection strategy that introduces imperceptible perturbations to prevent unauthorized learning. However, most existing UE methods produce perturbations strongly tied to specific training sets, leading to a significant drop in unlearnability when applied to unseen data or tasks. In this paper, we argue that for broad applicability, UEs should maintain their effectiveness across diverse application scenarios. To this end, we conduct the first comprehensive study on the transferability of UEs across diverse and practical yet demanding settings. Specifically, we identify key scenarios that pose significant challenges for existing UE methods, including varying styles, out-of-distribution classes, resolutions, and architectures. Moreover, we propose **Versatile Transferable Generator** (VTG), a transferable generator designed to safeguard data across various conditions. Specifically, VTG integrates Adversarial Domain Augmentation (ADA) into the generator's training process to synthesize out-of-distribution samples, thereby improving its generalizability to unseen scenarios. Furthermore, we propose a Perturbation-Label Coupling (PLC) mechanism that leverages contrastive learning to directly align perturbations with class labels. This approach reduces the generator's reliance on data semantics, allowing VTG to produce unlearnable perturbations in a distribution-agnostic manner. Extensive experiments demonstrate the effectiveness and broad applicability of our approach. Code is available at https://github.com/zhli-cs/VTG.

## 1   Introduction

The abundance of free internet data has facilitated the construction of numerous datasets, significantly advancing the field of deep learning [1, 2, 3, 4]. Despite their pivotal roles, growing concerns exist over the potential unauthorized exploitation of online personal data. To safeguard individual privacy, Unlearnable Examples (UEs) [5, 6, 7] have gained prominence in recent years. Models trained on unlearnable data tend to learn the correspondence between labels and perturbations instead of real semantics, resulting in poor classification accuracy on clean test images.

However, most existing UE methods generate perturbations that are highly dependent on specific training data, resulting in a significant performance drop when applied to unseen data or tasks. For instance, several initial works [5, 8, 9, 10] only achieve unlearnability when both training and

---

*Equal contribution. The author order was determined by a coin flip.
†Corresponding authors

39th Conference on Neural Information Processing Systems (NeurIPS 2025).

testing data are drawn from the same distribution. While some recent studies [7, 11, 12] aim to improve the transferability of UEs across different datasets (e.g., CIFAR-10 [13] to SVHN [14]), their effectiveness remains substantially limited in real-world scenarios, such as when generalizing to unseen classes [7] or transferring across images with different resolutions [12].

In this work, we argue that for UEs to be practically useful, they must retain their effectiveness beyond their original training scenarios. To this end, we introduce the first comprehensive transferable evaluation framework, which identifies four progressively challenging UE scenarios: **Intra-Domain**, **Cross-Domain**, **Cross-Task**, and **Cross-Space**. Specifically, **Intra-Domain** represents the conventional setting, where the training and test data are drawn from the same distribution. **Cross-Domain** extends this by considering cases where the training and test sets share the same classes but originate from different distributions, such as images with different styles in domain adaptation [15, 16, 17, 18, 19, 20, 21]. **Cross-Task** further increases the challenge by introducing both distribution shifts and class mismatches. For example, applying UEs generated from CIFAR-10 to SVHN. Finally, **Cross-Space** is the most challenging scenario, where even the input space differs between training and test sets, such as transferring UEs across images of different resolutions. In addition, following [5, 8, 10, 22, 23], we also consider the **Cross-Architecture** scenario, where we evaluate the generalizability of UEs across different network architectures.

Table 1: Comparative evaluation of existing UE methods across diverse scenarios. Symbols "✓", "✗", and "×" indicate "capable", "capable but ineffective" and "incapable", respectively.

| Method | Intra-Domain | Cross-Domain | Cross-Task | Cross-Space | Cross-Architecture |
|---|---|---|---|---|---|
| EMN [5] | ✓ | × | × | × | ✗ |
| LSP [11] | ✓ | × | × | × | ✓ |
| TUE [7] | ✓ | ✗ | × | × | ✗ |
| GUE [8] | ✓ | ✗ | × | × | ✓ |
| 14A [12] | ✓ | ✗ | ✗ | × | ✓ |
| **Ours** | ✓ | ✓ | ✓ | ✓ | ✓ |

To enhance the transferability of unlearnable perturbations across the aforementioned scenarios, we propose **Versatile Transferable Generator** (VTG), a generator designed to improve the generalization of unlearnability beyond specific domains or tasks. Unlike conventional methods that produce perturbations tightly coupled with the training data, VTG incorporates Adversarial Domain Augmentation (ADA) during training by synthesizing diverse worst-case out-of-distribution samples. Through this process, VTG compels the generator to learn perturbations that remain effective beyond a fixed data distribution and is iteratively refined to produce more transferable UEs that prevent the surrogate model from capturing meaningful semantics. As a result, the generator becomes less dependent on data-specific representations, improving its transferability to unseen conditions. Moreover, we also introduce Perturbation-Label Coupling (PLC) to further enhance VTG's transferability by directly aligning perturbations with class labels. Specifically, we leverage the surrogate model and CLIP text encoder to encode perturbations and labels, aligning perturbation embeddings with their corresponding label embeddings through a contrastive learning paradigm. Notably, this alignment is performed without relying on data semantics, enabling the generator to produce unlearnable perturbations in a distribution-agnostic manner. As a result, VTG achieves improved generalization across diverse data distributions, further strengthening its resilience to out-of-distribution variations. We note that 14A [12] also employs a generator architecture to address the transferability issue. However, its effectiveness is not consistently satisfactory across comprehensive scenarios.

Table 1 compares VTG with various state-of-the-art methods in terms of transferability across different scenarios. As shown, only VTG is capable of generating transferable UEs that remain effective in all settings, which will be further validated through empirical results in Section 4.

Our main contributions are summarized as follows:

- We introduce the first comprehensive evaluation framework to analyze the transferability of UEs across diverse practical scenarios, including Intra-Domain, Cross-Domain, Cross-Task, Cross-Space, and Cross-Architecture.

- We propose VTG, a versatile transferable generator effective across diverse scenarios. VTG introduces Adversarial Domain Augmentation to generate diversified samples and compel the generator to produce perturbations beyond fixed distributions. Moreover, the Perturbation-Label Coupling mechanism employs contrastive learning to align perturbations with class labels, introducing unlearnability in a distribution-agnostic manner.

- We empirically validate the efficacy of our method within the proposed comprehensive transferable setting. Extensive experiments demonstrate VTG's superior performance and broad applicability across diverse scenarios.

## 2 Related Work

Unlearnable examples [5] refer to a type of data that hinders deep learning models from acquiring informative knowledge. Models trained on such examples often achieve high accuracy on the training set but exhibit significantly degraded performance on the clean test set. This task is crucial to improving the security of machine learning, with applications in image classification [22, 24, 25, 26, 27, 28] and segmentation [29]. Implementation strategies for unlearnable examples can be broadly categorized into three paradigms. The first paradigm concentrates on injecting perturbations into data and creating "shortcuts" to minimize the classification loss [5, 8, 11]. The goal is to trick models into learning correspondences between perturbations and labels instead of acquiring real semantics. The second paradigm involves the strategic introduction of perturbations to deceive classifiers into associating images with incorrect classes [12, 30, 31]. Consequently, models trained on perturbed data tend to align images with false classes, thereby enhancing data protection. The last paradigm aims to produce perturbations without the utilization of surrogate models [11, 32, 33], which crafts unlearnable perturbations that are extremely easy to classify, such as utilizing the linearly-separable property or employing random convolution filters to convolve images of different classes.

The transferability of unlearnable examples has attracted some researchers, particularly concerning their generalization across different classes [7, 12] and architectures [5, 8]. However, related research remains relatively scarce, highlighting a critical gap in the literature. In this work, we introduce a comprehensive transferable study for unlearnable examples. We then propose a versatile perturbation generator to craft transferable unlearnable examples based on adversarial domain augmentation and perturbation-label coupling. Our approach effectively facilitates the transferability of unlearnable examples across diverse practical scenarios, thereby enhancing privacy protection in various real-world machine learning applications.

## 3 Proposed Method

In this section, we first present our comprehensive transfer-focused UE evaluation setting. Then, we introduce our Versatile Transferable Generator in detail.

### 3.1 Transferable Unlearnable Examples

Let $(\boldsymbol{x}, y)$ be a labeled example, where $\boldsymbol{x} \in \mathcal{X}$ is a data point, and $y \in \mathcal{Y} = \{1, \ldots, K\}$ is its label. We denote the source training set, the target training set, and the target test set as $\mathcal{D}_{source}$, $\mathcal{D}_{target}$, and $\mathcal{D}_{test}$, respectively. A VTG $\mathcal{G}$, trained exclusively on $\mathcal{D}_{source}$, is applied to $\mathcal{D}_{target}$ to generate an unlearnable version $\mathcal{D}'_{target}$ by introducing perturbations to each example $\boldsymbol{x}' = \boldsymbol{x} + \boldsymbol{\delta}$, where $\boldsymbol{\delta} = \mathcal{G}(x)$ represents either sample-wise or class-wise perturbations. A classifier $f_\lambda$, parameterized by $\lambda$, is then trained on $\mathcal{D}'_{target}$, and the unlearnability of $\boldsymbol{\delta}$ is evaluated by measuring the performance degradation of $f_\lambda$ on $\mathcal{D}_{test}$. In this work, we aim to conduct a comprehensive study on the transferability of UEs across different scenarios. Therefore, $\mathcal{D}_{source}$ can differ from $\mathcal{D}_{target}$ (and $\mathcal{D}_{test}$) in various ways, such as different distributions (Cross-Domain), label space $\mathcal{Y}$ (Cross-Task), or even input space $\mathcal{X}$ (Cross-Space).

Our comprehensive transfer-focused evaluation setting is composed of five distinct scenarios: **Intra-Domain**, **Cross-Domain**, **Cross-Task**, **Cross-Space**, and **Cross-Architecture**. In the **Intra-Domain** scenario, the original training dataset is partitioned into two subsets: a source training set and a target training set. The distributions of these subsets remain identical, ensuring they belong to the same domain. The **Cross-Domain** scenario represents a more challenging setting, where the source and target datasets share identical class labels but differ in their underlying data distributions, such as variations in style across paintings, photographs, and sketches. Moreover, in the **Cross-Task** scenario, the target dataset diverges from the source dataset not only in distributions but also in class labels, introducing a shift in task objectives. Furthermore, the **Cross-Space** scenario is the most complex scenario, where even the input space of the target set is different from the source set, such as differences in image resolution. Lastly, to be consistent with prior work [5, 8], the **Cross-Architecture** scenario assesses transferability when the classifiers employed for the source and target datasets differ in architecture.

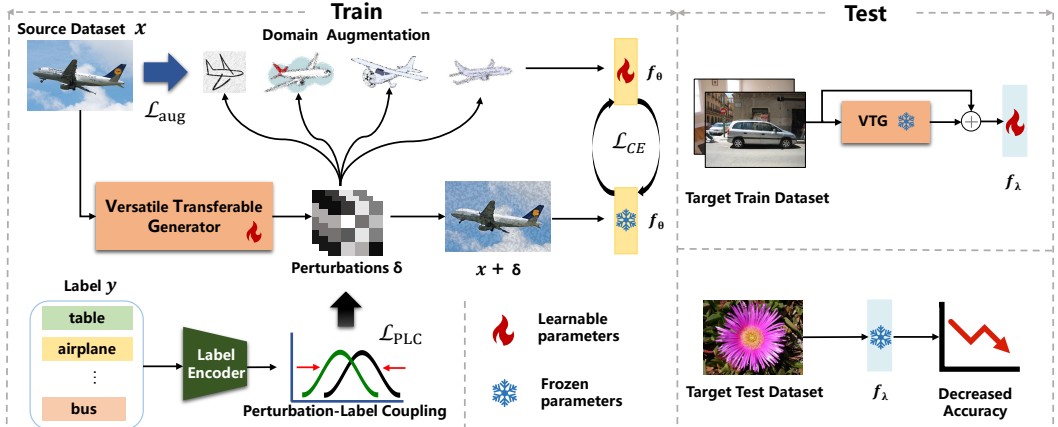

Figure 1: Overall pipeline of the proposed approach. Our Versatile Transferable Generator (VTG) is first optimized on the source dataset. Specifically, we leverage adversarial domain augmentation to diversify training samples, thereby requiring VTG to generate unlearnable perturbations capable of confusing a more generalizable classifier $f_\theta$. Then, we incorporate a perturbation-label coupling mechanism that enhances distribution-agnostic transferability by aligning perturbations with class labels. On the target dataset, we initially train a classifier $f_\lambda$ on the target training dataset that has been perturbed by the VTG-generated perturbations, followed by validating its decreased accuracy on the clean target test dataset.

## 3.2 Versatile Perturbation Generator

**Overview.** To generate perturbations that can generalize to various conditions, we propose a versatile unlearnable example generator capable of introducing unlearnability under various real-world scenarios. As shown in Figure 1, our VTG is primarily optimized through an alternating min-min optimization framework, where Adversarial Domain Augmentation and CLIP-guided Perturbation-Label Coupling are incorporated to equip the perturbation generator with transferability to diverse distribution variations.

In the first minimization loop, the Adversarial Domain Augmentation procedure synthesizes diversified images outside original distributions to enrich the surrogate classifier with superior generalizability. During this stage, the surrogate classifier $f_\theta$ is optimized by minimizing the classification loss while the perturbation generator $\mathcal{G}$ remains frozen. In the second minimization loop, the perturbation generator produces perturbations to craft unlearnable examples, which creates shortcuts and misleads the learning process of the surrogate classifier, preventing it from learning real semantics. Meanwhile, the Perturbation-Label Coupling mechanism conducts cross-modal alignment between perturbations and labels to enhance distribution-agnostic transferability. In this stage, the perturbation generator $\mathcal{G}$ is optimized with the classification loss while the surrogate classifier $f_\theta$ is frozen.

**Adversarial Domain Augmentation.** Existing UE methods generate unlearnable examples tied to specific training datasets [5, 8, 34], which lack explicit transferable designs to promote the unlearnability in comprehensive real-world scenarios. To address this issue, we draw insight from domain augmentation [15, 16, 35, 36, 37, 38, 39] and learn a domain composer $\mathcal{C}_\mu$ with parameters $\mu$ to diversify training samples. Specifically, the domain composer generates novel domains lying outside the original distribution. Then, we integrate both the source domain and the generated out-of-distribution domain to train the surrogate classifier. We assume that if our perturbation generator can effectively create shortcuts for the more generalizable surrogate classifier, it can craft unlearnable examples that take effect across diverse scenarios.

Our domain composer has a standard encoder-decoder structure [40]. Given source images $\boldsymbol{x}$ as input, the domain composer $\mathcal{C}_\mu$ aims to synthesize novel domains that lie outside the original distribution, thereby simulating diverse scenarios. To accomplish this goal, we propose to maximize the divergence between the source distribution $\mathbb{P}_a$ and the generated pseudo-novel distribution $\mathbb{P}_b$, with the Wasserstein distance $\mathcal{W}_d$, also known as Optimal Transport (OT) distance, as metric:

$$\mathcal{W}_d\left(\mathbb{P}_a, \mathbb{P}_b\right) = \inf_{\pi \in \Pi(\mathbb{P}_a, \mathbb{P}_b)} \mathbb{E}_{\boldsymbol{x}_a, \boldsymbol{x}_b \sim \pi}\left[d\left(\boldsymbol{x}_a, \boldsymbol{x}_b\right)\right], \tag{1}$$

$$d\left(\boldsymbol{x}_a, \boldsymbol{x}_b\right) = 1 - \frac{z_\theta\left(\boldsymbol{x}_a\right)^\top z_\theta\left(\boldsymbol{x}_b\right)}{\left\|z_\theta\left(\boldsymbol{x}_a\right)\right\|_2 \left\|z_\theta\left(\boldsymbol{x}_b\right)\right\|_2}, \tag{2}$$

where $\Pi\left(\mathbb{P}_a, \mathbb{P}_b\right)$ represents the set of all distributions $\pi(\boldsymbol{x}_a, \boldsymbol{x}_b)$, $z_\theta$ denotes the feature extractor within classifier $f_\theta$, $\boldsymbol{x}_a$ and $\boldsymbol{x}_b$ indicate the original image samples and the generated samples, respectively, $d(\cdot, \cdot)$ denotes the cost function defined by cosine distance.

The Wasserstein metric can capture high-order distributional characteristics and measure the minimal cost required to transform the source distribution $\mathbb{P}_a$ into the novel distribution $\mathbb{P}_b$, as demonstrated in diffusion models [41] and domain adaptation [42]. We employ the Sinkhorn algorithm [43] to reduce the computational complexity, as detailed below.

$$\mathcal{W}_d(\cdot, \cdot) = \inf_{M \in \mathcal{M}} \sum_{i,j} [M \odot D]_{i,j}, \tag{3}$$

where the soft-matching matrix $M$ represents the joint distribution $\pi$ in Eq. 1, $D$ denotes the pairwise cosine distance matrix calculated by Eq. 2.

The domain composer $\mathcal{C}_\mu$ is optimized by maximizing the divergence measure between source images and the synthesized images as:

$$\arg\max_\mu \mathcal{L}_{\text{aug}} = \mathcal{W}_d\left(\mathcal{C}_\mu\left(\boldsymbol{x}\right), \boldsymbol{x}\right), \tag{4}$$

where $\mu$ denotes the parameters of the domain composer.

After obtaining the generated diversified images, we employ the cross-entropy loss $\mathcal{L}_{\text{CE}}$ to optimize the domain composer $\mathcal{C}_\mu$ and the surrogate classifier $f_\theta$ simultaneously, setting constraints for the domain augmentation procedure and ensuring the classifier's predictions are aligned with true labels.

$$\arg\min_{\theta, \mu} [\mathcal{L}_{\text{CE}}\left(f_\theta\left(\boldsymbol{x} + \boldsymbol{\delta}\right), y\right) + \mathcal{L}_{\text{CE}}\left(f_\theta\left(\mathcal{C}_\mu\left(\boldsymbol{x}\right) + \boldsymbol{\delta}\right), y\right)], \tag{5}$$

where $\boldsymbol{\delta}$ is initialized to zero at the beginning of training.

**Perturbation Generator.** Different from gradient-based UE approaches [5, 6] that inherently depend on specific training samples to generate fixed perturbations, we develop a transferable generator that can produce unlearnable perturbations for any image in a single forward pass, exhibiting superior generalizability and applicability for practical use.

Our perturbation generator $\mathcal{G}$ employs an encoder-decoder structure [40]. Given the clean image $\boldsymbol{x}_i$, the generator $\mathcal{G}$ produces unlearnable noise as $\boldsymbol{\delta} = \mathcal{G}(\boldsymbol{x}_i)$ and renders data unlearnable as $\boldsymbol{x}_i' = \boldsymbol{x}_i + \boldsymbol{\delta}$. The goal is to create shortcuts by minimizing the classification loss, thereby misleading the surrogate classifier's training process and associating labels with perturbations instead of true semantics. We employ the cross-entropy loss $\mathcal{L}_{\text{CE}}$ as the classification loss and update the generator accordingly:

$$\arg\min_{\boldsymbol{\delta}} \mathbb{E}_{(\boldsymbol{x}, y) \sim \mathcal{D}_{source}} \mathcal{L}_{\text{CE}}(f_\theta(\boldsymbol{x} + \boldsymbol{\delta}), y), \tag{6}$$

where $f_\theta$ denotes the surrogate classifier that remains frozen at this stage. Meanwhile, the perturbation generator $\mathcal{G}$ is optimized to minimize the classification loss $\mathcal{L}_{\text{CE}}$.

To assure the imperceptibility of perturbations, we propose to constrain the magnitude of the noise $\boldsymbol{\delta}$ via a soft hinge loss on the $L_\infty$ norm, which is defined as follows:

$$\mathcal{L}_{\text{hinge}} = \mathbb{E}_{\boldsymbol{x}} \max\left(0, \|\boldsymbol{\delta}\|_\infty - \epsilon\right), \tag{7}$$

where $\epsilon$ denotes the perturbation bound and is set to $8/255$, in accordance with prior works [5, 8, 12].

**Perturbation-Label Coupling.** This procedure is designed to optimize our generator to produce unlearnable perturbations in a distribution-agnostic manner. To achieve this, we aim to establish a strong correlation between perturbations and labels. Specifically, we utilize the surrogate model to extract perturbation embeddings and leverage CLIP's fixed text encoder $\mathcal{T}$ [44] to extract label embeddings. Then, we conduct cross-modal alignments [45, 46] to associate perturbations with labels through a contrastive learning framework. This process guides the generator to directly associate perturbations with labels without relying on the underlying image semantics. Consequently, our VTG can produce unlearnable perturbations that remain effective under distribution shifts and resolution variations, ensuring effectiveness across diverse practical scenarios.

Given generated perturbations $\boldsymbol{\delta}$ and text labels $l$, we formulate the perturbation-label coupling procedure as follows:

$$P = z_\theta(\boldsymbol{\delta}); L = \mathcal{T}(l), \tag{8}$$

where $P \in \mathbb{R}^{B \times Q}$ and $L \in \mathbb{R}^{K \times Q}$ denote the extracted perturbation embeddings and label embeddings, respectively, $B$ indicates the number of data samples in a mini-batch, $K$ denotes the number of classes in the source dataset, $Q$ signifies the dimensionality of the feature space, The functions $z_\theta$ and $\mathcal{T}$ denote the feature extractor of the surrogate model $f_\theta$ and CLIP's text encoder, respectively. We keep both of them frozen at this stage to specifically optimize the perturbation generator.

We employ contrastive learning [47] to align perturbations with text labels. The pairwise similarity matrix $S$ is calculated as $S = P \cdot L^T$. Then, we formulate the PLC loss as follows:

$$\mathcal{L}_{\text{PLC}} = -\frac{1}{B} \sum_{i=1}^{B} \log \left( \frac{\exp(S_{i,y_i})}{\sum_{k=1}^{K} \exp(S_{i,k})} \right), \tag{9}$$

where $B$ indicates the size of each mini-batch.

After PLC, perturbations crafted by the generator directly align with CLIP-encoded label embeddings, yielding distribution-agnostic unlearnability and thus enhancing the applicability across diverse distributions. Moreover, benefitting from CLIP's shared image-text space, where label embeddings are closely associated with its extensive pre-trained image corpus, promoting alignments between perturbations and text labels implicitly enriches VTG with semantic diversity and establishes latent connections to unseen target images. Consequently, the perturbation generator maintains unlearnability even when encountering semantically related but previously unseen classes.

**Training Process.** Generally, the optimization of our Versatile Transferable Generator follows an alternating min-min optimization scheme. During the first minimization loop, the domain composer $\mathcal{C}_\mu$ and the surrogate classifier $f_\theta$ are optimized to generate diversified images and minimize the classification loss. During the second minimization loop, the perturbation generator $\mathcal{G}$ is optimized to also minimize the classification loss. The complete training procedure is delineated in Algorithm 1.

---

**Algorithm 1** Transferable Unlearnable Noise Generation

---

**Require:** Surrogate model $f_\theta$, feature extractor $z_\theta$, domain composer $\mathcal{C}_\mu$, CLIP text encoder $\mathcal{T}$, training data $(\boldsymbol{x}, y) \in \mathcal{D}_{source}$, class label $l$, perturbation $\boldsymbol{\delta}$, perturbation bound $\epsilon$, first loop training steps $T$, training epochs $E$;
**Ensure:** Transferable perturbation generator $\mathcal{G}$;
 1: **for** $e = 1$ to $E$ **do**
 2:     **for** $t = 1$ to $T$ **do**
 3:         $i, \boldsymbol{x}_i, y_i = \text{Next}(\boldsymbol{x}, y)$;
 4:         Input $\boldsymbol{x}_i$ to $\mathcal{C}_\mu$ to calculate $\mathcal{L}_{\text{aug}}$;
 5:         Update $\mathcal{C}_\mu$ by maximizing $\mathcal{L}_{\text{aug}}$;
 6:         Input $\boldsymbol{x}_i$, $\mathcal{C}_\mu(\boldsymbol{x}_i)$ and $\boldsymbol{\delta}$ to $f_\theta$ to calculate $\mathcal{L}_{\text{CE}}$;
 7:         Update $\mathcal{C}_\mu$ and $f_\theta$ by minimizing $\mathcal{L}_{\text{CE}}$;
 8:     **end for**
 9:     **for** $\boldsymbol{x}_j, y_j$ in $\boldsymbol{x}, y$ **do**
10:         Input $\boldsymbol{x}_j$ to $\mathcal{G}$ and get $\boldsymbol{\delta} = \mathcal{G}(\boldsymbol{x}_j)$;
11:         Obtain unlearnable example $\boldsymbol{x}_j' = \boldsymbol{x}_j + \boldsymbol{\delta}$;
12:         Input $\boldsymbol{x}_j'$ to $f_\theta$ to calculate $\mathcal{L}_{\text{CE}}$;
13:         Input $\boldsymbol{\delta}$ to $z_\theta$ and get embedding $P = z_\theta(\boldsymbol{\delta})$;
14:         Input $l$ to $\mathcal{T}$ and get embedding $L = \mathcal{T}(l)$;
15:         Calculate $\mathcal{L}_{\text{PLC}}$;
16:         Calculate $\mathcal{L}_{\text{hinge}}$;
17:         Update $\mathcal{G}$ by minimizing $\mathcal{L}_{\text{CE}}$, $\mathcal{L}_{\text{PLC}}$, $\mathcal{L}_{\text{hinge}}$;
18:     **end for**
19: **end for**
20: **return** Transferable perturbation generator $\mathcal{G}$

---

$$\arg\min_{\theta,\mu} \mathbb{E}_{(\boldsymbol{x},y) \sim \mathcal{D}_{source}} \left[ \min_{\boldsymbol{\delta}} \mathcal{L}_{\text{full}} \left( f_\theta(\boldsymbol{x} + \boldsymbol{\delta}), y \right) \right], \tag{10}$$

$$\text{s.t.} \quad \|\boldsymbol{\delta}\|_\infty \leq \epsilon$$

$$\mathcal{L}_{\text{full}} = \mathcal{L}_{\text{aug}} + \mathcal{L}_{\text{CE}} + \mathcal{L}_{\text{PLC}} + \mathcal{L}_{\text{hinge}}, \tag{11}$$

where $\mathcal{L}_{\text{full}}$ represents the final training objective.

# 4 Experiments

In this section, based on the comprehensive transferable UE evaluation framework introduced in Section 3.1, we rigorously evaluate our Versatile Transferable Generator across a variety of real-world scenarios. Moreover, we evaluate our method against several defense strategies. Finally, we conduct ablation studies to demonstrate the role of individual components in our training framework.

### 4.1 Experimental Settings

**Datasets and Models.** We evaluate VTG on CIFAR-10 [13], CIFAR-100 [13], SVHN [14] and PACS [17]. The first three datasets have been widely used in the UE literature, while PACS serves as a standard benchmark for evaluating domain shifts and consists of four domains: Art Painting, Cartoon, Photo, and Sketch. If not specified otherwise, we use ResNet-18 [48] as the surrogate classifier and target model both in training and testing. We utilize the classification accuracy on the clean test set as the evaluation metric, where lower accuracy indicates stronger unlearnability and protectiveness.

**Implementation Details.** Our model is developed with the PyTorch framework and trained on a single RTX A5000 GPU. Consistent hyperparameter settings are employed across all experiments. We employ the Adam optimizer [50] with an initial learning rate of 0.001. We train our model for 30 epochs on CIFAR-10 and SVHN, and 50 epochs on CIFAR-100 and PACS. We set the first loop training step $T$ to 10 in all experiments. The input image resolution is standardized to $224 \times 224$ for PACS, while $32 \times 32$ for the remaining datasets. Perturbations are crafted in a class-wise manner, where we first generate perturbations for each sample, then we average the perturbations for each class. To ensure imperceptibility, we set the perturbation bound $\epsilon$ to $8/255$. Results are averaged over five runs with different random seeds.

Table 2: Test accuracy under the **Intra-Domain** and **Cross-Task** scenarios, with CIFAR-10, CIFAR-100, and SVHN as target datasets.

| Source | Method | CIFAR-10 | CIFAR-100 | SVHN |
|---|---|---|---|---|
| | Clean | 94.66 | 76.27 | 96.05 |
| | Random | 95.57 | 71.19 | 25.11 |
| CIFAR-10 | EMN [5] | 10.16 | 21.80 | 24.72 |
| | LSP [11] | 13.54 | 9.35 | **7.77** |
| | REM [6] | 15.18 | 69.26 | 95.98 |
| | TUE [7] | 10.03 | 5.10 | 12.93 |
| | GUE [8] | 13.25 | 3.87 | 8.17 |
| | 14A [12] | 41.34 | 17.47 | 83.87 |
| | PUE [49] | 10.62 | 8.46 | 12.01 |
| | **Ours (ResNet)** | 9.99 | **0.99** | 9.65 |
| | **Ours (ViT)** | **9.54** | 1.21 | 7.94 |
| CIFAR-100 | EMN [5] | 27.27 | 3.95 | 9.64 |
| | LSP [11] | 24.16 | 9.00 | 17.03 |
| | REM [6] | 93.94 | 1.89 | 95.97 |
| | TUE [7] | 94.31 | 1.21 | 96.02 |
| | GUE [8] | 94.28 | 8.35 | 95.87 |
| | 14A [12] | 40.02 | 17.36 | 85.18 |
| | PUE [49] | 11.61 | 2.62 | 18.58 |
| | **Ours (ResNet)** | **9.85** | 1.14 | 11.07 |
| | **Ours (ViT)** | 11.40 | **1.09** | **9.39** |
| SVHN | EMN [5] | 14.31 | 6.25 | 9.05 |
| | LSP [11] | 38.50 | 38.51 | 8.00 |
| | REM [6] | 94.26 | 69.97 | 49.01 |
| | TUE [7] | 93.91 | 69.42 | 9.12 |
| | GUE [8] | 94.31 | 48.37 | 13.70 |
| | 14A [12] | 39.23 | 15.69 | 83.59 |
| | PUE [49] | 11.40 | 6.04 | 14.21 |
| | **Ours (ResNet)** | **10.66** | 1.76 | **6.38** |
| | **Ours (ViT)** | 11.16 | **1.65** | 7.41 |

### 4.2 Main Results

**Intra-Domain Transferability.** We first evaluate the transferability of VTG in the intra-domain scenario. Specifically, we randomly extract $50\%$ data from the original training dataset as the source training set and utilize the remaining $50\%$ data as the target training set. Then, we utilize VTG to generate perturbations to make the target training set unlearnable and evaluate its impact on the clean test set. As shown in Table 2, our VTG demonstrates superior performance compared with baseline methods on CIFAR-10, CIFAR-100 and SVHN, achieving $9.54\%$, $1.09\%$ and $6.38\%$ test accuracy, which approximates random guessing. Given that our method is evaluated under the more challenging intra-domain scenario, its exceptional unlearnability and protective effects are further verified.

**Cross-Domain Transferability.** To evaluate the effectiveness of VTG in transferring to images with different domains, we conduct experiments on the PACS dataset with the leave-one-domain-out setting, where the perturbation generator is trained on three domains and evaluated on the remaining domain. As detailed in Table 3, we select EMN [5], LSP [11], TUE [7], GUE [8] and 14A [12] as representative methods. Our VTG demonstrates clear effectiveness in transferring unlearnability across different domains, outperforming GUE by $25.73\%$ and 14A by $14.25\%$

Table 3: Test accuracy results under the **Cross-Domain** scenario on PACS.

| Method | Art | Cartoon | Photo | Sketch | Avg. |
|---|---|---|---|---|---|
| Clean | 76.92 | 81.25 | 83.75 | 85.42 | 81.84 |
| Random | 54.33 | 76.79 | 76.88 | 81.77 | 72.44 |
| EMN [5] | 43.75 | 74.11 | 71.88 | 14.58 | 51.08 |
| LSP [11] | 49.48 | 59.81 | 65.62 | 80.99 | 63.98 |
| TUE [7] | 38.71 | 72.05 | 62.50 | **9.11** | 45.59 |
| GUE [8] | 42.71 | 32.81 | 67.19 | 26.56 | 42.32 |
| 14A [12] | 27.20 | 29.91 | 45.51 | 20.72 | 30.84 |
| **Ours (ResNet)** | 21.63 | 18.30 | **10.00** | 16.41 | **16.59** |
| **Ours (ViT)** | **20.31** | **15.18** | 17.19 | 20.57 | 18.31 |

in average performance. These results highlight the strong cross-domain transferability of our method, effectively injecting unlearnability into images regardless of their diverse variations in visual styles.

Table 4: Test accuracy under the high-to-low **Cross-Space** scenario.

| Source | Method | CIFAR-10 | CIFAR-100 | SVHN |
|---|---|---|---|---|
| Art | LSP[11] | 94.16 | 70.49 | 9.23 |
| | TUE[7] | 94.06 | 69.76 | 95.45 |
| | GUE[8] | 91.78 | 39.82 | 92.06 |
| | 14A[12] | 40.13 | 17.62 | 86.50 |
| | **Ours (ResNet)** | **10.88** | **1.20** | **7.26** |
| | **Ours (ViT)** | 11.12 | 1.67 | 15.04 |
| Cartoon | LSP[11] | 93.92 | 70.72 | **8.35** |
| | TUE[7] | 93.75 | 70.75 | 95.79 |
| | GUE[8] | 87.50 | 49.94 | 94.89 |
| | 14A[12] | 38.89 | 17.65 | 83.68 |
| | **Ours (ResNet)** | **9.45** | **1.69** | 15.94 |
| | **Ours (ViT)** | 10.04 | 3.78 | 10.21 |
| Photo | LSP[11] | 94.01 | 69.86 | 13.20 |
| | TUE[7] | 94.00 | 70.01 | 95.87 |
| | GUE[8] | 93.53 | 28.75 | 94.32 |
| | 14A[12] | 40.95 | 16.50 | 84.12 |
| | **Ours (ResNet)** | 10.03 | **1.01** | **8.52** |
| | **Ours (ViT)** | **9.46** | 1.70 | 11.06 |
| Sketch | LSP[11] | 93.30 | 70.15 | 10.79 |
| | TUE[7] | 94.25 | 70.36 | 95.94 |
| | GUE[8] | 81.42 | 42.28 | 92.93 |
| | 14A[12] | 35.22 | 15.95 | 85.25 |
| | **Ours (ResNet)** | 10.04 | **1.18** | **9.69** |
| | **Ours (ViT)** | **10.00** | 2.16 | 12.65 |

Table 5: Test accuracy under the low-to-high **Cross-Space** scenario.

| Source | Method | Art | Cartoon | Photo | Sketch |
|---|---|---|---|---|---|
| CIFAR-10 | LSP[11] | 54.69 | 38.02 | 64.06 | 25.00 |
| | TUE[7] | 47.92 | 76.04 | 69.53 | 82.81 |
| | GUE[8] | 50.48 | 27.68 | 66.88 | 15.36 |
| | 14A[12] | 40.87 | 75.95 | 66.67 | 68.84 |
| | **Ours (ResNet)** | **11.98** | **10.71** | 11.25 | **4.69** |
| | **Ours (ViT)** | 15.38 | 20.26 | **10.94** | 17.97 |
| CIFAR-100 | LSP[11] | 48.96 | 70.83 | 70.31 | 18.23 |
| | TUE[7] | 43.75 | 69.79 | 64.84 | 82.03 |
| | GUE[8] | 56.73 | 39.29 | 78.75 | 4.43 |
| | 14A[12] | 38.46 | 73.00 | 68.42 | 70.35 |
| | **Ours (ResNet)** | **13.94** | **11.16** | 14.37 | **2.34** |
| | **Ours (ViT)** | 14.09 | 15.71 | **11.18** | 10.55 |
| SVHN | LSP[11] | 45.83 | 52.08 | 69.53 | 21.09 |
| | TUE[7] | 31.77 | 72.40 | 69.53 | 82.81 |
| | GUE[8] | 49.04 | 20.09 | 66.25 | **2.60** |
| | 14A[12] | 36.54 | 73.84 | 67.25 | 64.32 |
| | **Ours (ResNet)** | **12.98** | 12.05 | 15.00 | 17.97 |
| | **Ours (ViT)** | 13.56 | **11.61** | **12.50** | 15.36 |

**Cross-Task Transferability.** To evaluate transferability, we conduct cross-task experiments where perturbations are generated on one dataset and tested on another with distribution and class shifts. As shown in Table 2, methods like TUE [7] and GUE [8] perform reasonably well when transferring from CIFAR-10, but struggle when the source is CIFAR-100 or SVHN. In contrast, our method consistently achieves state-of-the-art results across most settings, effectively preventing semantic learning. Even in the challenging CIFAR-10 to SVHN case, VTG reduces accuracy to near random-guessing, demonstrating strong unlearnability.

**Cross-Space Transferability.** To evaluate cross-space transferability, we conduct experiments between two dataset groups differing in domain, class, and resolution. The low-resolution group $(32 \times 32)$ includes CIFAR-10, CIFAR-100, and SVHN, while the high-resolution group is PACS $(224 \times 224)$. As shown in Tables 4 and 5, existing methods degrade significantly under resolution shifts, whereas VTG remains consistently effective in both high-to-low and low-to-high transfers. Notably, in the CIFAR-10 to Art setting, our method outperforms the second-best by nearly 30%, demonstrating its ability to generate robust unlearnable perturbations across diverse input spaces.

**Cross-Architecture Transferability.** We further assess the transferability of VTG across network architectures. Using ResNet-18 as the surrogate model, we evaluate its ability to render other models unlearnable, including VGG16 [51], ResNet-50 [48], DenseNet-121 [52], and ViT [53]. As shown in Table 6, VTG consistently induces unlearnability across all targets, outperforming prior UE methods with an average 3% lower test accuracy. These results highlight VTG's strong cross-architecture transferability and practical effectiveness.

Table 6: Test accuracy under the **Cross-Architecture** scenario on CIFAR-10, where ResNet-18 is used as the surrogate model.

| Method | Network Architecture | | | |
|---|---|---|---|---|
| | VGG16 | ResNet-50 | DenseNet-121 | ViT |
| EMN [5] | 29.30 | 17.90 | 18.60 | 24.37 |
| DC [22] | 25.35 | 20.56 | 21.44 | 28.05 |
| CG [23] | — | 11.30 | 13.40 | — |
| SG [10] | 12.32 | 17.35 | 16.59 | 10.64 |
| GUE [8] | 13.72 | 12.97 | 13.71 | 16.77 |
| **Ours** | **8.92** | **10.03** | **9.69** | **10.53** |

## 4.3 Resistance to Defense Strategies

To assess the efficacy of our method against defense strategies, we train models on perturbed data with data transformation (such as Cutout [55], CutMix [56], and Mixup [57]), Adversarial Training [58], and UE-targeted defenses including D-VAE [59] and AN-SDA [60]. As shown in Table 7, existing methods exhibit substantial degradation in the unlearnable effect under defense measures. Two model-free approaches, AR [33] and OPS [54], show partial robustness against data transformation or adversarial training, but neither maintains unlearnability against UE-targeted defenses. In contrast, our method employs Adversarial Domain Augmentation to strengthen the perturbation generator against diverse image transformations. Moreover, the Perturbation-Label Coupling mechanism enables VTG

Table 7: Test accuracy of ResNet-18 against defense under the **Intra-Domain** scenario, where ResNet-18 is used as the surrogate model. "AT" denotes Adversarial Training.

| Method | w/o | Cutout | CutMix | Mixup | AT | D-VAE | AN-SDA |
|---|---|---|---|---|---|---|---|
| Clean | 94.66 | 95.10 | 95.50 | 95.01 | 84.99 | 93.29 | 92.76 |
| NTGA [9] | 42.46 | 42.07 | 27.16 | 43.03 | 70.05 | 89.21 | 89.00 |
| EMN [5] | 10.16 | 20.63 | 26.19 | 32.83 | 84.80 | 91.42 | 88.01 |
| REM [6] | 15.18 | 26.54 | 29.02 | 34.48 | 47.51 | 86.38 | 79.28 |
| SG [10] | 24.42 | 24.12 | 29.46 | 39.66 | 76.38 | 38.89 | 59.80 |
| LSP[11] | 13.54 | 19.87 | 20.89 | 26.99 | 84.59 | 91.20 | 64.34 |
| AR[33] | 11.75 | 12.36 | 18.02 | 14.59 | 83.17 | 91.77 | 80.20 |
| OPS[54] | 15.56 | 61.68 | 76.40 | 33.13 | 11.08 | 88.95 | 78.83 |
| **Ours** | **9.99** | **10.03** | **14.11** | **13.71** | **10.83** | **10.57** | **28.27** |

Table 8: Test accuracy of ResNet-18 with ImageNet* as the source dataset. Results of 14A on Flowers, Cars, and Food are cited from the original paper, with models trained on the full ImageNet.

| Source | Method | CIFAR-10 | CIFAR-100 | SVHN | Art | Cartoon | Photo | Sketch | Flowers | Cars | Food |
|---|---|---|---|---|---|---|---|---|---|---|---|
| | Clean | 94.66 | 76.27 | 96.05 | 76.92 | 81.25 | 83.75 | 85.42 | 84.47 | 40.43 | 65.45 |
| ImageNet* | LSP [11] | 29.04 | 11.32 | 8.90 | 28.12 | 74.48 | 74.22 | 79.95 | 10.13 | 1.95 | **1.16** |
| | 14A [12] | 39.90 | 11.40 | 80.38 | 35.10 | 69.62 | 67.25 | 66.83 | 15.15 | 6.69 | 16.01 |
| | **Ours** | **15.04** | **5.68** | **7.60** | **27.60** | **24.11** | **12.50** | **15.10** | **9.28** | **1.93** | 9.34 |

to produce distribution-agnostic perturbations with reduced reliance on image semantics. As a result, VTG ensures unlearnability while providing robust protection against various defense strategies.

## 4.4 More Comparison on ImageNet

To further evaluate the transferability of VTG with SOTA UE studies, such as the data-free LSP [11] and the transfer-oriented 14A [12], we utilize the ImageNet dataset [1] as the source dataset and assess the effectiveness of perturbations on downstream datasets. Specifically, we randomly select a subset from the first 100 classes of ImageNet to construct a smaller ImageNet*. The downstream datasets include CIFAR-10, CIFAR-100 [13], SVHN [14], PACS [17], Flowers [61], Cars [62], and Food [63], where the last three datasets are in accordance with previous work [12]. As presented in Table 8, our method outperforms competing approaches across most target datasets. This superior performance not only underscores the strong transferability but also highlights its remarkable adaptability, thereby demonstrating its potential for reliable deployment in diverse and challenging environments.

## 4.5 Effectiveness of Different Components

Our framework incorporates several key components to facilitate the transferability of UEs, including a vanilla Perturbation Generator, Adversarial Domain Augmentation, and Perturbation-Label Coupling. In this section, we meticulously assess the effect of these components in rendering data unlearnable. Table 9 illustrates the contributions of these components in both intra-domain and cross-task scenarios. Specifically, Adversarial Domain Augmentation improves transferability by exposing the surrogate model to diverse out-of-distribution samples, enabling the generator to craft more generalizable perturbations. Meanwhile, the Perturbation-Label Coupling mechanism reduces the generator's reliance on image semantics by aligning perturbations with text labels. The synergistic interaction between these two components substantially enhances UE's transferability, underscoring their essential role in our framework.

## 4.6 Further Analyses

**Visualizations of Perturbed Images.** In the task of Unlearnable Examples, the imperceptibility of added perturbations to human eyes serves as a critical evaluation criterion, and we specifically incorporate a soft hinge loss to constrain the norm of the generated perturbations. To provide a more vivid demonstration, we utilize the CIFAR-10 dataset [13] as the source dataset to train our VTG, then transfer it across various datasets and visualize the crafted unlearnable examples. The target datasets consist of CIFAR-10 [13], CIFAR-100 [13], and PACS [17]. As illustrated in Figure 2, the results demonstrate that the unlearnable examples generated by our VTG maintain good quality and the perturbations are basically invisible to humans.

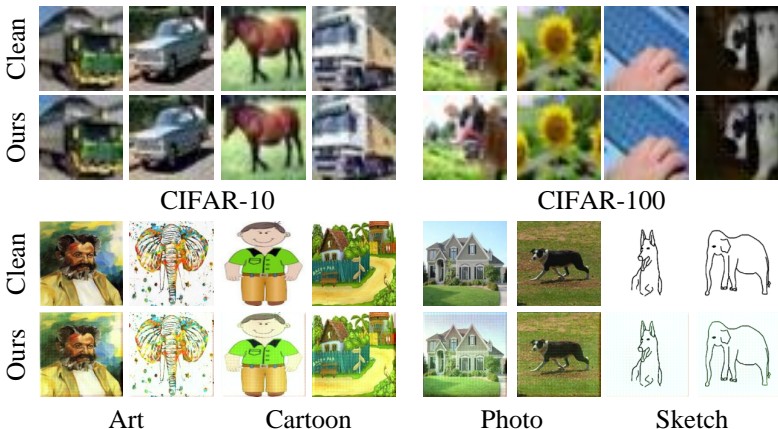

Figure 2: Visualization of clean images and unlearnable examples generated by VTG. The bottom two rows display results from PACS, which consists of four styles: Art, Cartoon, Photo, and Sketch.

Table 9: Ablation study on individual components of our versatile transferable generator. "PG", "ADA", and "PLC" denote the vanilla Perturbation Generator, Adversarial Domain Augmentation, and Perturbation-Label Coupling, respectively. All variants are trained on CIFAR-10 and evaluated under the **Intra-Domain** and **Cross-Task** scenarios with ResNet-18.

| Variant | PG | ADA | PLC | CIFAR-10 | CIFAR-100 | SVHN |
|---------|----|----|----|----------|-----------|------|
| 1 | ✓ | | | 16.53 | 10.19 | 19.59 |
| 2 | ✓ | ✓ | | 10.92 | 3.36 | 15.94 |
| 3 | ✓ | | ✓ | 12.89 | 5.73 | 11.05 |
| **Ours** | ✓ | ✓ | ✓ | **9.64** | **0.99** | **9.65** |

**Inference Cost Comparison.** To evaluate the practical applicability of VTG across various scenarios, we measure the inference cost and compare it with several generator-based approaches. Specifically, we feed each generator with a total of 3,200 images of size $224 \times 224$, using a batch size of 32. To ensure reliable and fair comparison, we compute the average inference time over five runs. All experiments are conducted on a consistent hardware setup comprising an Intel(R) Xeon(R) Silver 4210R CPU and an RTX A5000 GPU. As summarized in Table 10, our VTG contains fewer parameters and achieves reduced inference time and computational cost, highlighting its efficiency and practicality for real-world deployment.

Table 10: Comparison of inference cost and generator parameters across different methods.

| Method | Time (ms/img) | GFLOPs | Parameters (M) |
|--------|---------------|--------|----------------|
| GUE[8] | 3.7 | 8927.76 | 7.79 |
| 14A[12] | 6.3 | 15506.28 | 121.13 |
| **Ours** | **0.4** | **868.99** | **0.09** |

## 5   Conclusion

In this paper, we introduce the first comprehensive transferable study on unlearnable examples, which encompasses five progressive scenarios: Intra-Domain, Cross-Domain, Cross-Task, Cross-Space, and Cross-Architecture. This study establishes a new standard to rigorously assess the generalization performance of UE methods. Moreover, we propose a novel Versatile Transferable Generator with specialized designs to facilitate the transferability across various scenarios. Specifically, VTG employs Adversarial Domain Augmentation and Perturbation-Label Coupling to promote a superior transferable unlearnable example generator. Extensive experiments demonstrate the remarkable performance of our method compared with other state-of-the-art methods.

**Limitations.** While our method demonstrates strong transferability and superior applicability in practice, it still requires the VTG generator to craft perturbations at test time. Compared to methods that store fixed precomputed noises (e.g., EMN and TUE), our approach incurs marginal computational overhead during deployment, which we have discussed in the Supplementary.

## Acknowledgements

Z. Li, J. Cai, G. Xu, C. Ling and B. Wang are supported by the Natural Sciences and Engineering Research Council of Canada (NSERC), Discovery Grants program. Q. Li, F. Zhou, and S. Yang are supported by the National Natural Science Foundation of China (No.52302487).

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

# Appendix

This appendix provides a comprehensive analysis and in-depth exploration of our proposed Versatile Transferable Generator (VTG).

## A  Experimental Details

### A.1  More Implementation Details

**Implementation of VTG.** For the perturbation generator and the domain composer in ADA, we employ the Adam optimizer [50] with an initial learning rate of 0.001. For different surrogate models, we employ the SGD [64] optimizer with an initial learning rate of 0.1 for ResNet-18 and 0.01 for ViT.

**Baseline Types.** We conducted a comprehensive comparison against baseline methods, including both class-wise and sample-wise perturbation approaches, as shown in Tables 2– 5 of our paper.

- We included comparisons with the class-wise version of EMN [5] and PUE [49].
- We compared TUE [7] and LSP [11], two sample-wise methods that exhibit class-wise characteristics in their implementations: TUE employs a Class-wise Separability Discriminant to produce transferable, linearly separable perturbations, while LSP exhibits class-wise clustering, indicating strong class-dependent structures.
- Our comparison also covered three strict sample-wise approaches, including REM [6], GUE [8], and 14A [12].

These comparisons enable a comprehensive and fair evaluation of VTG against both class-wise and sample-wise baselines, where VTG consistently achieves superior performance in various scenarios.

**Implementation of Baselines.** Since non-generator-based baselines (e.g., LSP [11], TUE [7], and REM [6]) generate fixed perturbations, we first resample these perturbations before applying them to the images to ensure a fair comparison.

(1) In the Cross-Task scenario, when the number of samples differs, we generate new perturbation samples using a uniform sampling strategy introduced by TUE [7], thereby ensuring compatibility with the target dataset.

- If more classes are required, we interpolate two classes to create new class-wise perturbations:
$$\delta^* = \alpha\delta_i + (1 - \alpha)\delta_j, \text{where } y_i \neq y_j. \tag{12}$$
- If more samples within one class are required, we interpolate two samples to create new sample-wise perturbations:
$$\delta^* = \alpha\delta_i + (1 - \alpha)\delta_j, \text{where } y_i = y_j. \tag{13}$$

Note that $\delta$ and $y$ denotes perturbations and class labels; The subscripts denote class indexes; the number of newly created perturbations is controlled by varying $\alpha$, which is generally set as 0.5.

(2) In the Cross-Space scenario, when a resolution mismatch occurs between the fixed perturbations and the target dataset, we resample the perturbations to match the target resolution.

### A.2  The Architecture of Perturbation Generator

In the main paper, we devise VTG to produce transferable perturbations and craft unlearnable examples. We denote our perturbation generator as $\mathcal{G}$, which employs a standard encoder-decoder architecture. This structure comprises three down-sampling convolution layers, four residual blocks [48], and three transposed convolution layers. The detailed architecture is shown in Table 11.

## B  Additional Experimental Results

### B.1  Further Analysis of Perturbations

We further conduct an in-depth analysis of the perturbations by examining their spatial dispersion and quantitative data quality, as summarized in Table 12. The Shannon entropy of the perturbations reflects

Table 11: The architecture of the perturbation generator.

| Block Name | Layer | Number |
|---|---|---|
| *Down-sampling layers* | | |
| Conv | Conv (3 × 3)
InstanceNorm
ReLU | × 3 |
| *Bottleneck layers* | | |
| Residual | ReflectionPad
Conv (3 × 3)
BatchNorm
ReLU
ReflectionPad
Conv (3 × 3)
BatchNorm | × 4 |
| *Up-sampling layers* | | |
| ConvTranspose | ConvTranspose (3 × 3)
InstanceNorm
ReLU | × 2 |
| ConvTranspose | ConvTranspose (6 × 6)
Tanh | × 1 |

their spatial dispersion, where higher entropy values indicate that perturbations are more uniformly distributed across spatial regions rather than concentrated in localized areas. Our perturbations achieve an average entropy of 6.02 on CIFAR-10, substantially higher than those of baseline methods. At the same time, VTG maintains comparable PSNR and SSIM values relative to other baselines, demonstrating high data quality. This globally distributed perturbation pattern mitigates reliance on the object shapes of individual samples and enhances the transferability of unlearnability across diverse scenarios.

Table 12: Quantitative comparison of perturbation properties and image quality.

| Metric | EMN | LSP | TUE | GUE | Ours |
|---|---|---|---|---|---|
| Shannon Entropy↑ | 1.47 | 2.22 | 3.13 | 1.96 | **6.02** |
| PSNR (dB) ↑ | 37.06 | 34.94 | 34.43 | **37.72** | 35.77 |
| SSIM ↑ | **0.9963** | 0.9930 | 0.9869 | 0.9960 | 0.9894 |

## B.2 Test Accuracy Curves

In this section, we display the test accuracy curves of ResNet-18 [48] models trained on the poisoned CIFAR-10 dataset under various unlearnable attacks, as shown in Figure 3. It can be observed that other UE methods, particularly EMN [5], exhibit an initial accuracy peak at the start of training. In this regard, hackers can employ early stopping to acquire semantics within the training data. Compared with EMN [5] and GUE [8], our method successfully injects unlearnability at the first epoch, which validates the efficacy of our perturbation generation approach. Additionally, our method exhibits superior unlearnable effects throughout the entire training process, further highlighting the effectiveness of our proposed VTG.

## B.3 Visualization

**T-SNE Visualization.** To illustrate the superior transferability of our VTG, we further present the t-SNE visualization [65], as shown in Figure 4. Our VTG exhibits notable transferability both in Intra-Domain and Cross-Task scenarios, which leads to the misclassification of clean test samples by the classifier. In contrast, while GUE successfully generates unlearnable perturbations within the CIFAR-10 Intra-Domain scenario, it fails to transfer the unlearnability of generated perturbations

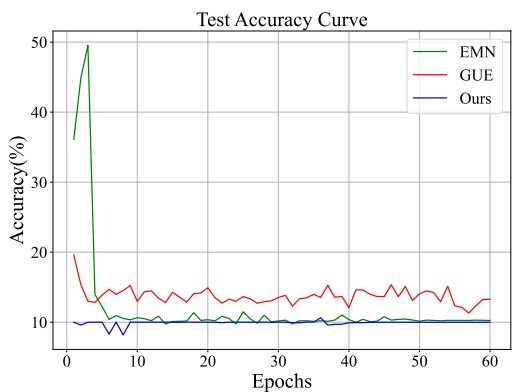

Figure 3: Test accuracy curves of ResNet-18 trained on poisoned CIFAR-10 under the **Intra-Domain** scenario with different unlearnable example methods.

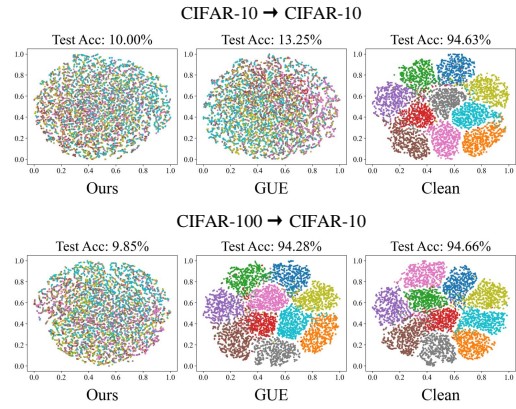

Figure 4: t-SNE visualization of classifier's last layer features, where classifiers are trained on the poisoned training set and tested on the clean test set. In the first row, perturbations are trained and tested on CIFAR-10. In the second row, perturbations are trained on CIFAR-100 and transferred to CIFAR-10.

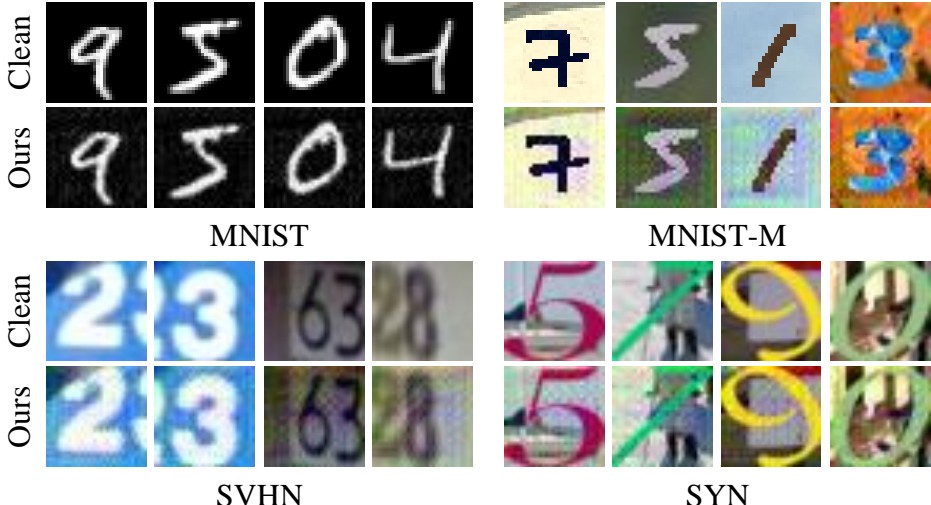

Figure 5: Visualization of clean images and unlearnable examples generated by VTG on MNIST, MNIST-M, SVHN, and SYN datasets.

from CIFAR-100 to CIFAR-10 in the Cross-Task scenario, thereby compromising the protection of real semantics in the target dataset.

**Perturbed Image Visualization.** Besides visualizations in Section 4.6, we provide more demonstrations on the Digits dataset, including MNIST [64], MNIST-M [66], SVHN [14], and SYN [66]. Similarly, we utilize the CIFAR-10 dataset [13] as a source to train VTG, then transfer it to target datasets and visualize the crafted unlearnable examples. As illustrated in Figure 5, the results demonstrate that the perturbations generated by our VTG are generally invisible to human eyes.

### B.4 Cross-Space Results on Digital Dataset

We further evaluate the transfer performance of VTG on Digital datasets, including MNIST [64], MNIST-M [66], and SYN [66]. As shown in Table 13, VTG consistently outperforms baseline methods and exhibits a substantial average improvement, further verifying its remarkable transferability.

Table 13: Test accuracy of ResNet-18 under the **Cross-Space** scenario, with Digits as target datasets.

| Source Dataset | Method | Digits | | | |
| | | MNIST | MNIST-M | SYN | Avg. |
| --- | --- | --- | --- | --- | --- |
| | Clean | 99.61 | 100.00 | 97.90 | 99.17 |
| | Random | 99.60 | 99.01 | 97.65 | 98.75 |
| CIFAR-10 | LSP [11] | 9.83 | **9.95** | 97.75 | 39.18 |
| | TUE [7] | 33.15 | 27.57 | 10.05 | 23.59 |
| | GUE [8] | **9.74** | 55.47 | 98.05 | 54.42 |
| | **Ours** | 9.93 | 10.15 | **9.65** | **9.91** |
| CIFAR-100 | LSP [11] | 9.80 | **9.28** | 96.25 | 38.44 |
| | TUE [7] | 33.69 | 38.75 | **10.15** | 27.53 |
| | GUE [8] | 21.75 | 99.02 | 97.95 | 72.91 |
| | **Ours** | **9.82** | 10.44 | 10.60 | **10.29** |
| SVHN | LSP [11] | 9.74 | 10.21 | 95.65 | 38.53 |
| | TUE [7] | **8.93** | 44.16 | 77.40 | 43.50 |
| | GUE [8] | 10.28 | 66.60 | 96.70 | 57.86 |
| | **Ours** | 10.28 | **9.74** | **40.45** | **22.92** |
| ImageNet* | 14A [12] | 97.05 | **30.06** | **79.70** | 68.94 |
| | **Ours** | **10.08** | 73.00 | 97.90 | **60.33** |

## B.5 More Analyses with Defenses

**Details of Defense Strategies.** We conduct a series of experiments to evaluate the effectiveness of VTG against different defense strategies. For data transformation, we employ Cutout [55], CutMix [56], and Mixup [57]. For adversarial training, which generally enhances robustness against adversarial perturbations [67], we utilize PGD-10 [58] with a poison radius of 4/255 and a step size of 2/255.

Table 14: Test accuracy of ResNet-18 under the **Intra-Domain** scenario with our VTG. Different defense strategies are applied, with "AT" denoting Adversarial Training.

| Dataset | w/o | Cutout | CutMix | Mixup | AT |
| --- | --- | --- | --- | --- | --- |
| CIFAR-100 | 1.14 | 1.29 | 1.20 | 2.40 | 1.98 |
| SVHN | 6.38 | 8.97 | 9.15 | 9.69 | 7.75 |

**Defenses under the Intra-Domain Scenario.** In our supplementary experiments, we evaluate the robustness of our approach under the Intra-Domain scenario using CIFAR-100 and SVHN as datasets. As shown in Table 14, our method consistently maintains a strong unlearnable effect across all defense strategies, effectively reducing model performance to levels approaching random guessing. This demonstrates the robustness and superiority of our approach in preserving unlearnability under varied Intra-Domain defense strategies.

**Defenses under the Cross-Task Scenario.** Moreover, we assess VTG's resilience to defenses under the Cross-Task scenario. As illustrated in Table 15, VTG consistently induces a significant unlearnability effect across diverse source-target pairs. For example, when CIFAR-10 is used as the source dataset and CIFAR-100 or SVHN as the target, VTG reduces test accuracy to extremely low levels, with similar patterns observed when CIFAR-100 and SVHN serve as sources. These findings substantiate that, even under the more demanding Cross-Task conditions, VTG effectively degrades model performance to levels approaching random guessing, thereby demonstrating its superior capacity to preserve unlearnability across diverse defense strategies.

## B.6 More Ablation Studies

**Ablation Studies with Adversarial Training.** To assess the contribution of each component of our VTG to its robustness against adversarial training, we conducted an ablation experiment on CIFAR-10.

Table 15: Test accuracy of ResNet-18 under the **Cross-Task** scenario with our VTG. Different defense strategies are applied, with "AT" denoting Adversarial Training.

| Source Dataset | Target Dataset | w/o | Cutout | CutMix | Mixup | AT |
|---|---|---|---|---|---|---|
| CIFAR-10 | CIFAR-100 | 0.99 | 0.93 | 1.28 | 1.47 | 1.33 |
| | SVHN | 9.70 | 15.88 | 9.92 | 6.28 | 9.18 |
| CIFAR-100 | CIFAR-10 | 9.85 | 10.14 | 10.09 | 13.14 | 11.86 |
| | SVHN | 11.07 | 15.94 | 8.72 | 19.59 | 16.08 |
| SVHN | CIFAR-10 | 10.66 | 10.43 | 9.52 | 69.70 | 89.08 |
| | CIFAR-100 | 1.76 | 41.81 | 10.38 | 34.13 | 55.01 |

Table 16 presents the results of this study. With only the baseline PG, the test accuracy is 13.27%, indicating a moderate unlearnability effect. The incorporation of ADA (Variant 2) improves this effect, reducing the accuracy to 10.17%, while the inclusion of PLC (Variant 3) yields an accuracy of 11.72%. Notably, when both ADA and PLC are integrated with PG (Ours), the test accuracy further decreases to 9.79%, demonstrating a synergistic improvement. These findings indicate that while the baseline generator provides some robustness against adversarial training, the additional components further enhance the unlearnability of the generated perturbations.

Table 16: Ablation study on the impact of individual components under adversarial training for our VTG. "PG", "ADA", and "PLC" denote the vanilla perturbation generator, Adversarial Domain Augmentation, and Perturbation-Label Coupling, respectively. All variants are trained on CIFAR-10 and evaluated under the **Intra-Domain** scenario with ResNet-18.

| Variant | PG | ADA | PLC | AT Radius | Accuracy |
|---|---|---|---|---|---|
| 1 | ✓ | | | 4 / 255 | 13.27 |
| 2 | ✓ | ✓ | | 4 / 255 | 10.17 |
| 3 | ✓ | | ✓ | 4 / 255 | 11.72 |
| **Ours** | ✓ | ✓ | ✓ | 4 / 255 | **9.79** |

**Ablation Studies on the Strength of Perturbations.** We conduct a series of ablation studies by varying the noise strength in the Intra-Domain scenario and the Cross-Task scenario, as shown in Table 17. Overall, unlearnability is approaching chance level as perturbation strength increases, while the imperceptibility to human eyes correspondingly degrades.

Table 17: Ablation study on perturbation strengths, with ResNet-18 as the surrogate model.

| Source Dataset | Target Dataset | 2 / 255 | 4 / 255 | 8 / 255 | 16 / 255 |
|---|---|---|---|---|---|
| CIFAR-10 | CIFAR-10 | 11.88 | 10.00 | 9.99 | 10.03 |
| CIFAR-10 | CIFAR-100 | 1.54 | 1.49 | 0.99 | 1.23 |
| CIFAR-10 | SVHN | 9.71 | 15.12 | 9.65 | 6.52 |

**Ablation Studies on ADA.** To further validate the role of ADA in boosting the transferability of perturbations, we replace ADA with commonly adopted data augmentation techniques, including Cutout [55], Cutmix [56], and Mixup [57]. As shown in Table 18, ADA consistently outperforms these standard augmentations, especially for cross-task transfer (e.g., CIFAR-10 to CIFAR-100), while other augmentation strategies perform poorly in this transfer scenario. This observation highlights the effectiveness of ADA in crafting transferable unlearnable examples across various scenarios.

**Ablation Studies on Wasserstein Distance.** We adopt the Wasserstein distance in ADA for its ability to capture high-order distributional characteristics. It is efficiently computed via the Sinkhorn algorithm with minimal overhead. In contrast, simple metrics (e.g., L2 distance) only capture point-wise discrepancies and mean shifts, overlooking high-order information essential for quantifying domain shifts. We report results in Table 19 and exclude PLC for better clarity.

**Ablation studies on the pre-trained text encoder of PLC.** To assess the sensitivity of our approach to text encoders pretrained on different datasets in PLC, we conduct ablation studies using several

Table 18: Ablation studies of ADA against standard data augmentation strategies under the Intra-Domain and Cross-Task scenarios, where ResNet-18 is used as the surrogate model.

| Source Dataset | Target Dataset | Cutout | Cutmix | Mixup | ADA (Ours) |
|---|---|---|---|---|---|
| CIFAR-10 | CIFAR-10 | 12.93 | 10.52 | 11.25 | **9.99** |
| CIFAR-10 | CIFAR-100 | 8.12 | 7.83 | 2.87 | **0.99** |
| CIFAR-10 | SVHN | 19.59 | 16.27 | 10.66 | **9.65** |
| CIFAR-100 | CIFAR-10 | 19.57 | 13.85 | 13.20 | **9.85** |
| CIFAR-100 | CIFAR-100 | 3.06 | 14.98 | 4.05 | **1.14** |
| CIFAR-100 | SVHN | 23.81 | 18.07 | 18.90 | **11.07** |
| SVHN | SVHN | 14.68 | 10.48 | 14.79 | **10.66** |
| SVHN | CIFAR-100 | 47.06 | 3.31 | 6.96 | **1.76** |
| SVHN | SVHN | 15.06 | 15.94 | 11.07 | **6.38** |

Table 19: Comparison of distance metrics in ADA, with ResNet-18 as the surrogate model.

| | Source Dataset | CIFAR-10 | CIFAR-100 | SVHN |
|---|---|---|---|---|
| wo ADA | CIFAR-10 | 16.53 | 10.19 | 9.91 |
| L2 Distance | CIFAR-10 | 14.33 | 4.56 | 9.93 |
| Wasserstein | CIFAR-10 | **10.92** | **3.36** | **7.07** |

CLIP variants trained on diverse corpora (YFCC-15M, CC12M, and LAION-400M). Additionally, we include the multi-modal model BLIP for further comparison. All models are kept fixed and are employed solely to extract label embeddings within the PLC mechanism. The results, summarized in Table 20, show that across all settings, the generated unlearnable examples consistently reduce model performance to near random-guessing levels. This indicates that our method is not dependent on a specific CLIP variant or pretraining corpus but instead exploits the general semantic alignment properties inherent to large-scale multi-modal pretraining. Consequently, the PLC mechanism is resistant to dataset-specific artifacts and biases arising from large-scale web pretraining.

Table 20: Impact of text encoders pretrained on different datasets in PLC on unlearnable performance.

| Label Encoder | CIFAR-10 → CIFAR-10 | CIFAR-10 → CIFAR-100 | CIFAR-10 → SVHN |
|---|---|---|---|
| CLIP (YFCC-15M) | 12.72 | 1.03 | 9.21 |
| CLIP (CC12M) | 10.02 | 1.71 | 8.62 |
| BLIP (14M) | 10.54 | 2.03 | 8.62 |
| CLIP (LAION-400M) | 9.99 | 0.99 | 9.65 |

**Ablation studies on PLC loss weightings.** We conduct ablation studies on different PLC loss weightings in the Cross-Space scenario, as shown in Table 21. The results show that removing the PLC loss leads to the worst performance, confirming its necessity. Meanwhile, varying the PLC weight has a relatively minor impact, indicating the stability of our method to this hyperparameter.

**Ablation studies on the depth of the generator.** To examine the effect of generator depth, we train four generator variants containing 2, 4, 8, and 16 residual blocks. The source dataset is CIFAR-10, and the target datasets are CIFAR-10, CIFAR-100, and SVHN. As shown in Table 22, all variants reduce test accuracy to near random-guessing levels, indicating that unlearnability performance is largely consistent across different generator depths.

## B.7 Impact of Training Ratio

In the context of Intra-Domain transferability, we split the original training dataset, designating a portion of samples as the source training dataset and the remaining samples as the target training set. Results presented in Table 23 demonstrate that the ratio of the source training dataset to the overall training dataset exerts an insignificant influence on the unlearnable effects of our VTG. Leveraging the collaborative effects of Adversarial Domain Augmentation and Perturbation-Label Coupling, our method can generate highly effective unlearnable perturbations even with limited training data, which

Table 21: Performance with varying PLC loss weightings under the **Cross-Space** scenario, with CIFAR-10 and PACS being utilized as the source and target dataset, respectively.

| PLC Weight | Source Dataset | Art | Cartoon | Photo | Sketch |
|---|---|---|---|---|---|
| 0 | CIFAR-10 | 22.60 | 22.32 | 16.88 | 20.27 |
| 1.0 | CIFAR-10 | 11.98 | **10.71** | 11.25 | 4.69 |
| 2.0 | CIFAR-10 | **9.13** | 18.75 | **10.00** | **4.43** |

Table 22: Effect of generator depth on unlearnable performance.

| Depth | CIFAR-10→CIFAR-10 | CIFAR-10→CIFAR100 | CIFAR-10→SVHN |
|---|---|---|---|
| 2 | 11.63 | 1.11 | 9.57 |
| 4 | 9.99 | 0.99 | 9.65 |
| 8 | 10.05 | 1.22 | 9.35 |
| 16 | 8.11 | 1.22 | 8.69 |

Table 23: Ablation study on the selected portion of the source dataset in the **Intra-Domain** scenario. A portion value of 0 indicates the use of random noise as input.

| Portion | 0 | 0.1 | 0.2 | 0.3 | 0.4 |
|---|---|---|---|---|---|
| Test accuracy | 95.57 | 10.09 | 10.03 | 10.01 | 9.99 |

| Portion | 0.5 | 0.6 | 0.7 | 0.8 | 0.9 |
|---|---|---|---|---|---|
| Test accuracy | 9.99 | 10.02 | 10.01 | 9.99 | 10.09 |

efficiently introduces unlearnability and prevents classification models from extracting meaningful semantics from the protected dataset.

## B.8 Training Time Comparison

Table 24: Training time comparison (s/epoch) with ResNet-18 as the surrogate model.

| Method | CIFAR10 Training Time | SVHN Training Time |
|---|---|---|
| EMN | 18.72 | 32.50 |
| TUE | 22.12 | 55.77 |
| GUE | 644.97 | 719.31 |
| 14A | 1158.09 | 1690.09 |
| **Ours** | 46.60 | 65.72 |

To evaluate the training cost in practical scenarios, we report the per-epoch training time on CIFAR-10 (50,000 samples) and SVHN (73,257 samples), comparing our method with both gradient-based baselines (EMN, TUE) and generator-based baselines (GUE, 14A). All experiments are conducted on a single NVIDIA RTX A5000 GPU. As shown in Table 24, gradient-based methods require less training time to obtain unlearnable perturbations; however, their applicability to new classes and unseen samples is limited. In contrast, generator-based methods incur higher training costs but are capable of crafting unlearnable examples for novel data, thereby exhibiting superior practical applicability. We observe that our method incurs a moderately higher training cost than EMN [5] and TUE [7], yet demonstrates greater transferability across diverse scenarios. Compared with GUE (which relies on implicit gradients) and 14A (which employs a much larger generator, 121.13M vs. 0.09M parameters), our approach achieves superior efficiency and transferability.

