# OpenReview forum: "Versatile Transferable Unlearnable Example Generator"
_NeurIPS.cc/2025/Conference — NeurIPS 2025 poster_

### Official Review · Reviewer_mhbh · 2025-06-28

**Clarity:** 2
**Significance:** 2
**Originality:** 3
**Rating:** 3
**Confidence:** 2

**Summary:**

This paper presents a method for generating “unlearnable” examples under feature or label distribution shifts. Unlearnable examples are data points that cause models trained on them to learn the correspondence between labels and perturbations instead of capturing the true semantics, leading to poor classification accuracy on clean test images. The paper proposes training a domain composer to generate out-of-distribution (OOD) data and a perturbation-label coupling mechanism to learn the perturbation generator in a nearly distribution-free setting. Across varying levels of distribution shift in labels or features, the authors demonstrate that their algorithm outperforms existing approaches.

**Questions:**

What's the difference between data poisoning attack that aims at decreasing test accuracy and generating unlearnable examples?

**Ethical Concerns:**

["NO or VERY MINOR ethics concerns only"]

**Final Justification:**

The authors study a setting where data must be released with visual similarity to the original points but rendered useless for training machine learning models. This goal differs from the standard notion of privacy protection, which typically aims to prevent leakage of private (e.g., visual) information about training data from model outputs.

Their results rely on the strong assumption that the defender can manipulate all examples in the training set. In scenarios where this is not possible, a model may memorize the “outlier” perturbed (unlearnable) data points, potentially increasing privacy leakage for those specific points. However, the results use test accuracy as the only metric, and does not consider the partial-perturbation setting, which makes it less clear whether the proposed method helps provide privacy.

The paper would also benefit from a clearer statement of the problem setting and motivation, as well as a precise definition of “unlearnable examples.”  I will keep my score and lean toward rejection. However, I am less confident in my assessment due to limited familiarity with this setting, and my main concerns focus on the privacy implications of unlearnable examples, which may be tangential to the paper’s primary contribution of providing a universal, distribution-free generator for unlearnable examples.

**Limitations:**

Yes

**Quality:**

2

**Strengths And Weaknesses:**

**Strengths**
I find this paper addresses an interesting problem by proposing a novel method and empirically demonstrating its superiority over existing approaches.

**Weaknesses**

- The definition of “unlearnable” in this context is somewhat unclear: success is measured by degraded test accuracy, which is similar to a specific type of clean-label attacks in the data-poisoning literature.
- Although the goal is to render individual data points “unexploitable/unlearnable” by deep models and thus protect their privacy, the privacy implications of training on these modified “unlearnable” examples remain unexplored. Prior work shows that poisoned examples often exhibit higher memorization scores [1], and because the perturbations here are small, extracting an “unlearnable” example may reveal the original data. In practice, this approach may even worsen privacy, if a company unknowingly trains on poisoned data.

[1] Tramèr et al. Truth Serum: Poisoning Machine Learning Models to Reveal Their Secrets. 2022.

---

> ### Author Rebuttal · Authors · 2025-07-31
>
> We thank reviewer mhbh for the positive feedback and valuable suggestions. Below, we provide responses to each weakness and question.
>
> **W1&Q1:** We would like to clarify the difference between data poisoning attacks and unlearnable examples from three aspects [1, 2, 3, 4].
>
> **(1) Different Goals**. Data poisoning is an **attack technique** that aims to manipulate model behavior of specific samples during inference (e.g., causing cats to be misclassified as dogs). In contrast, unlearnable examples are a **defense technique** designed to protect privacy. They introduce imperceptible perturbations, encouraging the model to **rely on perturbations for classification rather than learning meaningful semantic features**. As a result, the model is learning nothing meaningful from unlearnable examples, effectively enhancing privacy protection.
>
> **(2) Different Methodological Designs.** Data poisoning attacks operate as an adversarial game, often employing strategies like feature collision or gradient matching to craft poisoned samples that deliberately collide with target class features. In contrast, **unlearnable examples aim at decouple model learning from real data semantics**, which establishes direct associations between perturbations and labels, instead of images and labels. Therefore, privacy is protected since models are not learning real semantics for classification but relying on perturbations.
>
> **(3) Different Perturbed Ratios for Decreasing Test Accuracy**. While data poisoning and unlearnable examples both decrease the test accuracy, data poisoning attacks typically inject **a small number** of imperceptibly poisoned samples into the training set, aiming to achieve poisoning effect (manipulate model prediction) and remain stealthy. In contrast, unlearnable examples aim at preventing **the entire dataset** from unauthorized exploitation. Therefore, it adds perturbations to each sample and tricks the model to learn perturbations instead of real semantics, leading to good privacy protection.
>
> **W2:** We clarify that unlearnable examples do not increase the risk of privacy leakage. To evaluate whether our method compromises data privacy, we conduct a standard **Membership Inference Attack (MIA) [5]**, which attempts to determine whether a given sample was part of the model’s training set based solely on its outputs. This attack is **widely used as a proxy for measuring privacy leakage**.
>
> We compare our method against representative UE baselines, including EMN, LSP, TUE, and GUE. Instead of reporting absolute attack accuracy that may vary with dataset and task, we report below each method's difference in MIA accuracy compared to an anchor model trained with clean data. All models are trained and evaluated on CIFAR-10 to ensure consistent and fair comparison.
>
> | Method       | EMN   | LSP   | TUE    | GUE    | Ours   |
> |:------------:|:-----:|:-----:|:------:|:------:|:------:|
> | ΔMIA vs. Clean ↓ | +0.06 | +0.02 | -4.48  | +6.94  | -8.24  |
>
> We observe that all evaluated UE methods, except GUE, either maintain MIA accuracy at clean baseline levels or further reduce privacy risks. Notably, **our method achieves the greatest reduction in MIA accuracy, indicating strong protection against training data extraction**. Contrary to concerns about data exposure, **our unlearnable perturbations actually enhance the model's resilience to membership inference attacks**.
>
> [1] Unlearnable Examples: Making Personal Data Unexploitable. ICLR 2021.
>
> [2] A Survey on Unlearnable Data. arXiv 2025.
>
> [3] Transferable Clean-Label Poisoning Attacks on Deep Neural Nets. ICML 2019.
>
> [4] Poison Frogs! Targeted Clean-Label Poisoning Attacks on Neural Networks. NeurIPS 2018.
>
> [5] Membership inference attacks against machine learning models. IEEE Symposium on Security and Privacy 2017.

---

> > ### Comment · Reviewer_mhbh · 2025-08-06
> >
> > Thank you for your response. It clarifies the distinction between poisoning attacks and unlearnable examples. I also appreciate the additional results on MIA accuracy. Are these averages taken over all data points?

---

> > > ### Author Response · Authors · 2025-08-06
> > >
> > > We appreciate the reviewer for the further comments. We’re glad to hear that our rebuttal helped clarify the distinction between poisoning attacks and unlearnable examples.
> > >
> > > We also confirm that the reported MIA accuracy is computed over **the entire CIFAR-10 dataset**, covering all training and test samples. This ensures a fair and comprehensive comparison. Furthermore, all baseline methods and our proposed approach are evaluated under the same protocol to maintain consistency across results.
> > >
> > > If you have any further questions or feedback, we would be happy to continue the discussion. Thank you again for your thoughtful review.

---

> > > ### Author Response · Authors · 2025-08-08
> > >
> > > We are glad that our rebuttal has helped clarify the distinction between poisoning attacks and unlearnable examples. If you have any further questions or concerns, we would be happy to continue the discussion.
> > >
> > > Given your acknowledgment of the clarifications and the additional MIA results, we would be truly grateful if you might consider reflecting this more explicitly in your final evaluation score. Your support would significantly contribute to highlighting the value of our contribution.
> > >
> > > Thank you again for your valuable feedback and consideration.

---

> > > > ### Comment · Area_Chair_8rNH · 2025-08-08
> > > >
> > > > Dear authors,
> > > >
> > > > Please refrain from explicitly pressuring reviewers to increase their scores.
> > > >
> > > > Best
> > > > AC

---

> > > > > ### Author Response · Authors · 2025-08-08
> > > > >
> > > > > Dear AC,
> > > > >
> > > > > Thank you for your kind reminder and guidance. We sincerely apologize and fully respect the policy regarding communication with reviewers. Our sole intention was only to express our willingness to address any remaining concerns before the rebuttal phase concludes.
> > > > >
> > > > > We greatly value the reviewer’s constructive and insightful feedback, which has been instrumental in improving the clarity and quality of our work, and we also sincerely appreciate your guidance in helping us maintain a constructive discussion process.
> > > > >
> > > > > We will not initiate any further follow-up messages.
> > > > >
> > > > >
> > > > > Best Regards,
> > > > >
> > > > > Submission 26468 Authors

---

### Official Review · Reviewer_bdeR · 2025-06-29

**Clarity:** 3
**Significance:** 3
**Originality:** 4
**Rating:** 4
**Confidence:** 5

**Summary:**

This paper presents a novel framework for generating Unlearnable Examples (UEs) that maintain their effectiveness across diverse machine learning scenarios. In contrast to prior methods, which often fail to generalize beyond the specific conditions of their training data, the proposed Versatile Transferable Generator (VTG) is designed to produce UEs that remain robust across varying tasks, domains, input resolutions, and model architectures. By leveraging two core components—Adversarial Domain Augmentation (ADA) and Perturbation-Label Coupling (PLC)—VTG demonstrates consistent superiority over existing UE approaches across five distinct transferability settings.

**Questions:**

- Since the method leverages the PLC mechanism, do the generated perturbations tend to resemble class-level patterns rather than being strictly sample-specific?
- Does the use of CLIP introduce additional vulnerabilities, given its potential biases and limitations inherited from pretraining on large-scale web data?
- Regarding the results in Table 7, were all defense strategies applied simultaneously during the generation of the perturbed data, or was each evaluated defense strategy applied individually?

**Ethical Concerns:**

["NO or VERY MINOR ethics concerns only"]

**Final Justification:**

After discussions with the authors, my primary concern remains the focus on class-level perturbations. As the authors themselves noted, prior work has shown that class-wise perturbations are easier to detect and recover, which may limit the stealthiness and effectiveness of the attack. Therefore, I will maintain my score as “Borderline Accept.”

**Limitations:**

- As stated in the paper, the proposed method introduces additional computational overhead.

**Quality:**

4

**Strengths And Weaknesses:**

Strengths:

- The paper is clearly organized and provides thorough explanations of the model architecture, loss functions, training methodology, and datasets used.
- The proposed method, VTG, is novel and offers a partial solution to a critical challenge in machine learning privacy—the transferability of unlearnable examples. Extensive experiments validate its superiority over existing baselines across five challenging real-world scenarios.
- The results demonstrate strong resilience against common defense strategies, including adversarial training, which has proven effective against most existing unlearnable example methods.


Weaknesses:

- The paper is lack of analyses. Although the results are compelling, there isn't enough theoretical nor empirical analyses such as visualizations to explain why the generated perturbations work.
- The paper lacks an ablation study. For example, it does not explore how the depth of the generator affects the effectiveness of the attack.
- The proposed method is compared to other sample-level approaches; however, since it incorporates the PLC mechanism, it raises the question of whether the generated perturbations are closer to class-level patterns. I would expect the authors to provide more clarification on this point.
- The PLC procedure relies on CLIP to extract text embeddings, which may inherit biases or vulnerabilities from its pretraining data.

---

> ### Author Rebuttal · Authors · 2025-07-31
>
> We thank reviewer bdeR for the positive feedback and valuable suggestions. Below we provide responses to each weakness and question.
>
> **W1:** To better explain why our generated perturbations are effective, we have presented **t-SNE visualizations** in the supplementary material and add **additional empirical analyses on the spatial dispersion of perturbations** in the rebuttal.
>
> **(1) T-SNE visualizations on classifier’s last layer features.** As shown in Supplementary Figure 3, we have presented t-SNE visualizations of model's learned feature representations on the clean test set, after training models on perturbed (unlearnable) data. We observe that GUE still yields well-separated class clusters when transferring perturbations from CIFAR-100 to CIFAR-10, which fails to prevent model from learning real semantics. In contrast, model trained on our unlearnable examples produce entangled and overlapping feature distributions on the clean test data, indicating that the model did not learn useful semantics from the perturbed training set.
>
> **(2) Perturbations' spatial dispersion.** Moreover, we assess the spatial dispersion of perturbations by computing the Shannon entropy of each perturbation map. A higher entropy reflects that the perturbation values are more evenly distributed across spatial locations, rather than being concentrated in localized areas. As shown below, our perturbations achieve an average entropy of 6.02 on CIFAR-10, markedly higher than other baseline methods while keeping comparable data quality (as the PSNR and SSIM metrics show below). **This global coverage reduces reliance on object shapes of specific samples and enhances the transferability of unlearnability across diverse scenarios.**
> |Metric | EMN   | LSP   | TUE   | GUE   | Ours  |
> | :---: |  :-:   | :-:   | :-:   | :-:   | :--:  |
> |Shannon Entropy of Perturbations ↑| 1.47 | 2.22 | 3.13 | 1.96 |**6.02** |
> |PSNR (dB) ↑| 37.06 | 34.94 | 34.43 | **37.72** | 35.77 |
> | SSIM ↑  |  **0.9963** | 0.9930 | 0.9869| 0.9960 | 0.9894 |
>
> **W2:** We add an ablation study on generator depth by training generators with 2, 4, 8, and 16 residual blocks on CIFAR-10, and evaluating their effectiveness on CIFAR-10, CIFAR-100, and SVHN. Results are shown below:
> | Depth  |  CIFAR-10→CIFAR-10 |  CIFAR-10→CIFAR100 |  CIFAR-10→SVHN |
> | :---------: | :-------: | :--------: | :----: |
> | 2           | 11.63   | 1.11     | 9.57 |
> | 4           | 9.99    | 0.99     | 9.65 |
> | 8           | 10.05   | 1.22     | 9.35 |
> | 16          | 8.11    | 1.22     | 8.69 |
>
> We observe that all variants achieve test accuracy close to the random guess level (10% for CIFAR-10/SVHN, 1% for CIFAR-100). **This indicates that the method is insensitive to the depths of the generator.**
>
> **W3&Q1:** **We clarify that our generated perturbations are indeed class-wise**. Specifically, we first generate perturbations for each sample, then we average the perturbations for each class and add the same perturbations for samples within the same class. This is also in accordance with the Perturbation-Label Coupling (PLC) mechanism that aligns perturbation embeddings with CLIP-encoded class-specific label embeddings. The class-wise perturbations are thus encouraged to cluster towards corresponding class labels, establishing direct and close associations between perturbations and labels.
>
> **W4&Q2:** We appreciate your insightful comment. While we acknowledge that pre-trained models may inherit biases or vulnerabilities from their training data, we believe the impact on our method is minimal. Specifically, **PLC does not use CLIP for prediction; it only leverages CLIP’s frozen text embeddings as a class-level prior to guide the clustering of class-wise perturbations**. This indirect integration limits the propagation of CLIP-specific biases into our model.
>
> To evaluate whether our approach is sensitive to potential biases from CLIP's pretraining data, we conduct ablation experiments using different CLIP checkpoints trained on different corpora (YFCC-15M, CC12M, and LAION-400M). We also include an alternative multi-modal model (BLIP). All models are solely used to extract label embeddings in the PLC procedure. The results are summarized below:
> | Label Encoder of PLC | CIFAR-10→CIFAR-10  | CIFAR-10→CIFAR-100 | CIFAR-10→SVHN   |
> | :-------------------: | :------: | :------: | :------: |
> |       CLIP (YFCC-15M)        |  12.72   |   1.03   |   9.21   |
> |         CLIP (CC12M)         |  10.02   |   1.71   |   8.62   |
> |         BLIP (14M)       |  10.54   |   2.03   |   8.62   |
> | CLIP (LAION-400M) | 9.99 | 0.99 | 9.65 |
>
> Across all settings, the generated unlearnable examples consistently reduce model performance to random guess level (10% for CIFAR-10/SVHN, 1% for CIFAR-100). This indicates that **our approach is not reliant on any specific CLIP variant or pretraining corpus, but instead leverages the general semantic alignment inherent in multimodal embeddings**. As a result, the PLC mechanism is robust to dataset-specific artifacts and potential biases arising from large-scale web-based pretraining data.
>
> **Q3:** We clarify that **we follow existing UE works [1, 2, 3] to evaluate each defense strategy individually**.  Defenses such as Cutout, CutMix, and Mixup are mutually exclusive by design, so we apply them independently.
>
> [1] Image Shortcut Squeezing: Countering Perturbative Availability Poisons with Compression. ICML 2023.
>
> [2] Purify unlearnable examples via rate-constrained variational autoencoders. ICML 2024.
>
> [3] Detection and defense of unlearnable examples. AAAI 2024.

---

> > ### Comment · Reviewer_bdeR · 2025-08-02
> >
> > I thank the authors for their response, which has addressed most of my concerns. However, the authors acknowledge that the generated perturbations are class-wise, which remains a primary issue. First, the paper only compares against sample-wise methods, making the comparison unfair. Second, class-wise perturbations are generally easier for defenders to detect than sample-wise ones, limiting the method's practical effectiveness. Therefore, I will maintain my score.

---

> > > ### Author Response · Authors · 2025-08-03
> > >
> > > We thank Reviewer bdeR for reading our rebuttal and for the prompt response. We are glad to see that most of the concerns have been addressed. For the remaining points, we would like to provide additional clarification.
> > >
> > > (1) We would like to clarify that we conducted a **comprehensive comparison** against six baseline methods, including both **class-wise** and **sample-wise** perturbation approaches, as shown in Tables 2–5 of our paper.
> > > - We included comparisons with the specific **class-wise version** of EMN.
> > > - We compared TUE and LSP, two sample-wise methods that exhibit **class-wise characteristics** in their implementations: TUE employs a **Class-wise Separability Discriminant** to produce transferable, linearly separable perturbations, while LSP exhibits **class-wise clustering**, indicating strong class-dependent structures.
> > > - Our comparison also covered three strict **sample-wise** approaches, including REM, GUE, and 14A.
> > >
> > > To further **enhance fairness**, we include two explicitly **class-wise** methods, CUDA [1] and PUE [2], **as additional baselines**. These comparisons enable a comprehensive and fair evaluation of our method against both class-wise and sample-wise strategies, and our method consistently achieves **superior performance** in various scenarios.
> > >
> > > | Method  |  CIFAR-10→CIFAR-10 |  CIFAR-10→CIFAR100 |  CIFAR-10→SVHN |
> > > | :---------: | :-------: | :--------: | :----: |
> > > | CUDA          | 18.48   | 27.77     | 65.72 |
> > > | PUE           | 10.62   |   8.46   | 12.01|
> > > | Ours          | **9.99**| **0.99** | **9.65** |
> > >
> > > (2) Regarding your second point, to the best of our knowledge, we are not aware of existing UE literature explicitly stating that class-wise perturbations are generally easier for defenders to detect than sample-wise perturbations. We would greatly appreciate it if you could kindly provide any relevant references on this. On the other hand, we recognize that some previous studies [3, 4] have suggested that class-wise perturbations might be more readily recovered. However, it remains unclear to us how such perturbations can be effectively recovered solely from the images after perturbation.
> > >
> > > We would be truly grateful for the opportunity to learn from your expertise on this matter and would greatly appreciate any guidance or suggestions you provide regarding practical approaches to achieving such detection or recovery.
> > >
> > > Additionally, in terms of practical performance, as you noted, our proposed method demonstrates strong resilience against common defense strategies compared to existing methods, including sample-wise unlearnable example algorithms. This resilience is detailed in Table 7 of our manuscript.
> > >
> > > Thank you once again for your valuable feedback.
> > >
> > > [1] CUDA: Convolution-based unlearnable datasets. CVPR 2023.
> > >
> > > [2] Provably Unlearnable Data Examples. NDSS 2025.
> > >
> > > [3] Autoregressive perturbations for data poisoning. NeurIPS 2022.
> > >
> > > [4] Game-theoretic unlearnable example generator. AAAI 2024.

---

### Official Review · Reviewer_Dthy · 2025-06-29

**Clarity:** 2
**Significance:** 3
**Originality:** 3
**Rating:** 5
**Confidence:** 3

**Summary:**

The paper studies the problem of generating unlearnable examples by perturbing the data. The idea is that we want to perturb inputs such that when you train a classifier on these inputs, it does not learn the semantics, and only learns the features of the perturbation. Compared to prior work, the authors propose a more rigorous evaluation framework where the perturbation generator is learned in one source task, and then applied to other target tasks that are unknown during training. They propose Versatile Transferable unlearnable example Generator (VTG), which is a complex method for training the perturbation generator. They show that VTG achieves superior unlearnability performance compared to prior state of the art in several of their evaluation settings.

**Questions:**

**Q1.** Please comment on W2.

**Q2.** Do I understand that the Adversarial Domain Augmentation could in principle be replaced with any other data augmentation strategy? If so, I would suggest to abstract it away in the presentation of the method, and present it after the core part of the method. I.e. you could just say that the first loop in your method is to train a classifier on the original data, with any strategy you may like. You could also ablate different kinds of augmentation strategies potentially (I am not requesting this for the rebuttal, just a suggestion).

**Q3.** What is Concat in Eq. (5)? Are you concating the images, or are you just aggregating the dataset across all augmentations? I think it's the latter, but then the notation is confusing.

**Q4.** Overall, it wasn't clear to me how the objectives (4) and (5) interact with each other? Do you first train against (4) for some number of steps and then train against (5)? Or do you train against a combination of both (4) and (5) simultaneously?

**Q5.** Could you present examples of what augmentations $C_\mu$ actually learns? I am not sure if you can put images in the rebuttal, but please include these in the updated versions of the paper, and describe in text in the rebuttal, if not.

**Q6.** I found the CLIP-based contrastive learning bit quite confusing. Is there any reason for the encodings of the classifier to be aligned with the text encoder of CLIP? In theory, they don't even need to be of the same dimension? My understanding is that $z_\theta$ is a part of the classifier $f_\theta$, and that it is frozen during the training on objective (9)? Overall, what is the motivation here, why do you want to use the image embedding?

**Q7.** In the experiments, are all the models (domain composer, classifier, perturbation generator) trained from scratch? Is CLIP text encoder the only pretrained model used?

**Q8.** At test time, is the perturbation generator $G$ only conditioned on the input image? It is not using the labels or any other information?

**Q9.** What are the limitations on the scalability of the method? Is it possible to apply it to the full ImageNet, as opposed to the first 100 classes? How does the runtime for training compare to the baselines? (not request for rebuttal, just a question)

**Q10.** When tuning the hyper-parameters such as the weights for the different losses, learning rates, etc, do you use the performance on the in-domain test set as a target?

**Ethical Concerns:**

["NO or VERY MINOR ethics concerns only"]

**Final Justification:**

The rebuttal addressed my concerns, and so I increase the score to 5. I encourage the authors to include all the new results and clarifications, as well as examples of generated images in the final version of the paper.

**Limitations:**

Limitations adequately addressed.

**Paper Formatting Concerns:**

No concerns.

**Quality:**

3

**Strengths And Weaknesses:**

For calibration,  I want to mention that I was previously not aware of the field of generating unlearnable perturbations, and so I am not familiar with the prior work in this field.

# Strengths

**S1.** The evaluation setting that the authors develop appears to be more realistic than the evaluations done in the previous literature. It seems important to evaluate unlearnability under diverse tasks.

**S2.** The authors do a careful evaluation with many baselines and evaluation settings.

**S3.** The authors perform an ablation of the various components of the method suggesting that the different components are indeed useful.

**S4.** The proposed method performs well in the evaluations presented by the authors, outperforming the baselines significantly in most of the distribution shift settings.

**S5.** The paper is generally well-written.

# Weaknesses

**W1.** The proposed VTG method is quite complicated, and it's hard to understand what exactly each component is doing. I will ask several questions about the method in the questions section. But overall, I feel like the presentation of the method does not do a good enough job of explaining what is the motivation behind each component and how all of them fit together.

**W2.** The only metric that is considered in evaluation is the test accuracy, which should be as low as possible. Generally, it is very easy to generate perturbations that lead to poor accuracy: $\delta(x) = G(x) = -x$. You would get exactly random guess performance on any data distribution by applying this perturbation. Consequently, what makes the task non-trivial is that there should be some constraint on what the perturbation should be. The authors mention that the perturbation should be imperceptible, which seems like a reasonable requirement: if you want to apply the perturbations to the images you upload to the internet, you don't want the images to be visibly affected. The authors do incorporate a hinge loss on the size of the perturbation in Eq. (7), towards this goal. However, the imperceptability of the perturbations is not a part of the evaluations, as far as I understand. In particular, if I understand correctly, the perturbations could actually be quite large, if the optimization pressure is insufficient to make them small. Moreover, it is unclear how the size of the perturbations compares to the other baselines considered.

Overall, only using low test accuracy as a measure of performance without any additional constraints on the perturbations seems wrong. Another trivial way to game this objective is to use a very bad classifier, or use inadequate hyper-parameters for the classifier. Possibly there should be constraints such as high train accuracy, and high accuracy when trained on clean data with the same hyper-parameters. Perhaps this last concern is partially evaluated by reusing the same architecture and hyper-parameters for the classifier across all baselines.

---

> ### Author Rebuttal · Authors · 2025-07-31
>
> We thank reviewer Dthy for the positive feedback and valuable suggestions. Below we provide responses to each weakness and question.
>
> **W1:** We will elaborate on the motivation behind each module and the joint roles below.
>
> **(1) Overall Motivation.** Existing UE methods generate perturbations that are highly dependent on specific training data, which results in substantial performance drops when applied to unseen data. To address this issue, we aim to generate unlearnable examples that remain effective across various scenarios. To do so, we design the Versatile Transferable Generator, and incorporate Adversarial Domain Augmentation and Perturbation-Label Coupling to improve transferability by enhancing effectiveness under distribution shifts and reducing reliance on specific data to craft shortcuts.
>
> **(2) Motivation of Adversarial Domain Augmentation.** ADA aims at synthesizing worst-case out-of-distribution samples, which compels the perturbation generator to craft perturbations that remain effective for a more robust surrogate model. As a result, the optimized perturbation generator possesses better transferability and is capable of introducing unlearnability under distribution shifts.
>
> **(3) Motivation of Perturbation-label Coupling.** PLC aims at reducing UE's reliance on specific data when introducing shortcuts, playing a collaborative role in transferring perturbations to unseen samples. Its direct perturbation-label alignment is achieved without relying on data semantics, which improves the effectiveness of the perturbation generator with unseen data.
>
> Overall, VTG is superior in transferability to diverse scenarios by **leveraging ADA's generalizability over distribution shifts and PLC's less reliance on specific data**. We will revise the motivation accordingly in the updated paper for better clarification.
>
> **W2&Q1:** We emphasize that our perturbations are visually imperceptible and do not compromise image quality, and we will make clarifications below.
>
> **(1) The imperceptibility of perturbations.** We consider it from two aspects.
> - **Qualitative Visualizations.** We have included qualitative visualizations of the perturbed images in Figure 3 of Supplementary Section 1.4, where we compared the perturbed images with clean images across seven datasets, including CIFAR-10, CIFAR-100, MNIST, MNIST-M, SVHN, SYN, and PACS. The quality of perturbed images is generally acceptable, especially on CIFAR-10 and CIFAR-100. Since we are not allowed to upload images, please see Figure 1 of [1] for visualizations of other UE methods.
> - **Quantitative Evaluations.** To further justify the visual quality of our method, we add evaluations between perturbed and clean images using two widely adopted metrics: PSNR (Pixel Fidelity) [2] and SSIM (Structural Similarity) [3]. We report the average values computed on 100 perturbed images on CIFAR10 below. Our perturbations are relatively comparable with existing UE methods, indicating that the perturbed images maintain high visual quality.
>
> |Method|EMN|LSP|TUE|GUE|Ours|
> |:-:|:-:|:-:|:-:|:-:|:-:|
> |PSNR (dB) ↑|37.06|34.94|34.43|**37.72**|35.77|
> |SSIM ↑|**0.9963**|0.9930|0.9869|0.9960|0.9894|
>
> **(2) The size of the perturbations compared to other baselines.** We follow the same perturbation size (8/255) as other UE methods [4, 5, 6] for fair comparison.
>
> **(3) Additional constraints besides low test accuracy.** We clarify that we follow the standard protocol and use the same architectures and hyper-parameters for the classifier as existing UE works [4, 5, 6]. The classifier has good test accuracy if trained on clean data (please see Table 2, Table 3, and Table 8 in the main paper). Moreover, when a classifier is trained on perturbed images and evaluated on clean test images, the training accuracy reaches nearly 100\%, showing the classifier has learned to classify training samples. However, when testing on clean test samples, its prediction approximates random guessing, indicating the drop in test performance is not due to underfitting but because it's learning shortcuts instead of real semantics.
>
> **Q2:** We would like to clarify from two aspects that the Adversarial Domain Augmentation is important for optimizing transferable perturbations and is not replaceable with some standard data augmentation.
>
> **(1) The core motivation of ADA.** ADA aims at mimicking worst-case scenarios across distribution shifts, so that perturbations that can mislead the more robust surrogate model optimized by ADA are able to remain effective across various scenarios.
>
> **(2) Empirical Justification.** As shown in the table below, ADA consistently outperforms standard augmentations, especially for cross-task transfer (e.g., CIFAR-10 to CIFAR-100), while other augmentation strategies perform poorly in this transfer scenario. This observation highlights the effectiveness of ADA in improving the transferability of unlearnable examples.
> |Source|Target|Cutout|Cutmix|Mixup|ADA (Ours)|
> |:-:|:-:|:-:|:-:|:-:|:-:|
> |CIFAR-10|CIFAR-10|12.93|10.52|11.25|**9.99**|
> ||CIFAR-100|8.12|7.83|2.87|**0.99**|
> ||SVHN|19.59|16.27| 10.66|**9.65**|
> |CIFAR-100|CIFAR-10|19.57|13.85|13.20| **9.85**|
> ||CIFAR-100|3.06|14.98|4.05| **1.14**|
> ||SVHN|23.81|18.07|18.90|**11.07**|
> |SVHN|CIFAR-10|14.68|10.48|14.79|**10.66**|
> ||CIFAR-100|47.06|3.31|6.96| **1.76**|
> ||SVHN|15.06|15.94|11.07|**6.38**|
>
> **Q3:** We clarify that in Eq. (5), Concat does refer to aggregating the original and augmented samples within each mini-batch. We revise Eq. (5) as follows for better clarification:
> $$\underset{\theta,\mu}{\arg \min}[ \mathcal{L}\_{\mathrm{CE}}\left(f\_{\theta}\left( x+\delta \right), y \right) +\mathcal{L}\_{\mathrm{CE}}\left(f\_{\theta}\left( C\_{\mu}\left(x\right)+\delta \right), y \right)],
> $$
>
> **Q4:** We note that **the two objectives are optimized sequentially** rather than jointly. We first use Eq. (4) to optimize the domain composer $C_\mu$ to generate diversified samples, and then use Eq. (5) to train the composer $C_\mu$ and the classifier $f_\theta$ with the cross-entropy loss, which prevents $C_\mu$ from over-distorting the images to other classes.
>
> **Q5:** Since we are unable to include images in the rebuttal, we clarify that the augmented samples generated by our domain composer exhibit similar visual diversity and semantic consistency as those shown in Figure 1. To support this, we compute PSNR, SSIM, and LPIPS (Perceptual Similarity) [7] over 100 randomly selected samples as below. Figure 1 examples are representative samples that contain desirable domain shifts, while Gaussian-blurred examples serve as a weaker baseline with fewer domain shifts.
>
> |Metric|Generated Examples|GaussianBlur Examples|Examples in Figure 1|
> |:-:|:-:|:-:|:-:|
> |PSNR(dB)|18.85|23.20|16.35|
> |SSIM|0.6039|0.7290|0.5651|
> |LPIPS|0.5131|0.3379|0.5303|
>
> These results show that the augmentations generated by the domain composer match the examples in Figure 1 in terms of fidelity, distortion, and semantic similarity, which successfully diversifies training samples to optimize more transferable perturbations.
>
> **Q6:** We clarify that we use the feature extractor $z_\theta$ within the classifier $f_\theta$ to encode perturbations instead of images. We also leverage CLIP text encoder to obtain label embeddings. Both the feature extractor $z_\theta$ and CLIP text encoder are kept frozen during optimization of Eq. (9). **The motivation behind PLC is to reduce reliance on specific samples when generating perturbations.** As a result, strong perturbation-label associations are established, enhancing the transferability of unlearnable examples.
>
> **Q7:** Yes. We clarify that the domain composer, the surrogate classifier, and the perturbation generator are all trained from scratch, while the CLIP text encoder is the only pretrained model and is kept frozen.
>
> **Q8:** Yes. We note that the perturbation generator only receives raw images as input and produces corresponding perturbations. There are no labels or other information.
>
> **Q9:** We clarify that the limitation on the scalability of our method mainly derives from the moderately increased training time compared to gradient-based methods.
>
> **For runtime comparison**, we report the per-epoch training time on CIFAR-10 (50000 samples) and SVHN (73257 samples), and compare it against gradient-based baselines (EMN, TUE) and generator-based baselines (GUE, 14A). All experiments are conducted on a single NVIDIA RTX A5000 GPU. While our method has a moderately higher training cost than EMN and TUE, it holds greater transferability across various scenarios. When compared with GUE (requiring implicit gradient) and 14A (using a much larger generator, 121.13M vs. our 0.09M), our method is superior in both efficiency and transferability.
>
> **Regarding full ImageNet**, our method is fully compatible with large-scale datasets like ImageNet. Due to time and computational constraints during the rebuttal period, we are currently unable to provide results using a full 1000-class ImageNet.
> |Method|CIFAR10 Training Time (s/epoch)|SVHN Training Time (s/epoch)|
> |:-:|:-:|:-:|
> |EMN|18.72|32.50|
> |TUE|22.12|55.77|
> |GUE|644.97|719.31|
> |14A|1158.09|1690.09|
> |Ours|46.60|65.72|
>
> **Q10:** Yes, we used performance on the Intra-Domain test set to tune the hyper-parameters. Since our VTG targets transferability, we also used evaluations on the Cross-Task and Cross-Space scenarios.
>
> [1] Detecting and Corrupting Convolution-based Unlearnable Examples. AAAI 2025.
>
> [2] Scope of validity of PSNR in image/video quality assessment. Electronics Letters 2008.
>
> [3] Image quality assessment: from error visibility to structural similarity. IEEE Transactions on Image Processing 2004.
>
> [4] Unlearnable Examples: Making Personal Data Unexploitable. ICLR 2021.
>
> [5] Transferable Unlearnable Examples. ICLR 2023.
>
> [6] Game-Theoretic Unlearnable Example Generator. AAAI 2024.
>
> [7] Investigating loss functions for extreme super-resolution. CVPRW 2020.

---

> > ### Comment · Reviewer_Dthy · 2025-08-05
> > **Response to the rebuttal**
> >
> > Dear authors, thank you for the detailed rebuttal. Based on your response, and after reading the other reviews, I increased my score. Please include all the new results and clarifications in the updated versions of the paper.

---

> > > ### Author Response · Authors · 2025-08-05
> > >
> > > We sincerely thank Reviewer Dthy for the thoughtful feedback and for taking the time to read our rebuttal. We truly appreciate your decision to raise the score, as well as your recognition of our clarifications and new results. We will make sure to incorporate all relevant updates into the revised version of the paper. Thank you again for your constructive and encouraging response.

---

### Official Review · Reviewer_pQ4b · 2025-06-30

**Clarity:** 3
**Significance:** 4
**Originality:** 3
**Rating:** 5
**Confidence:** 5

**Summary:**

This paper aims to improve the transferability of unlearnable examples across various scenarios. It combines Adversarial Domain Augmentations and the Perturbation-Label Coupling mechanism to train a Versatile Transferable Generator. The trained generator generalizes very well to unseen data from multiple domains.

**Questions:**

How do you generate EMN and TUE perturbations for the target training data since both of them are neither generator-based methods like GUE, 14A, nor label-determined methods like LSP?

**Ethical Concerns:**

["NO or VERY MINOR ethics concerns only"]

**Final Justification:**

I maintained my positive score after the rebuttal.

**Limitations:**

yes

**Paper Formatting Concerns:**

No paper formatting concerns

**Quality:**

4

**Strengths And Weaknesses:**

Strengths:

1. This paper systematically benchmarks the transferability of unlearnable examples across multiple scenarios, including intra-domain, cross-domain, cross-task, cross-space, and cross-architecture, which is very important for applying UEs as a practical real-world data protection tool.

2. It introduces a novel component, the domain composer, into the UE area, which leads to the enhancement of the domain transferability.

3. The results in Table 8 are impressive, as they show that a generator trained on ImageNet can serve as a "foundation" model to execute protections for different domains simultaneously.

4. Experiments are comprehensive. The structure is well-organized. It's easy to read.


Weakness:

1. It's unclear why aligning the perturbation feature extracted by a surrogate model with the label feature extracted by the CLIP text encoder can lead to Perturbation-Label Coulpling. This process seems to make perturbations in the same class cluster around a certain point.  I don’t see the CLIP text encoder playing an irreplaceable role here.

2. This paper did not present augmented images after the domain composer $C_\mu$. I'm curious about how much the composer distorts the clean images and if it works precisely as the illustration (various planes) in Figure 1 shows.

3. The defense part missed diffusion-based purification methods (refer to [1,2,3]).
	[1] Diffusion models for adversarial purification.
	[2] The Devil's Advocate: Shattering the Illusion of Unexploitable Data using Diffusion Models.
        [3] BridgePure: Limited Protection Leakage Can Break Black-Box Data Protection.

---

> ### Author Rebuttal · Authors · 2025-07-31
>
> We thank reviewer pQ4b for the positive feedback and valuable suggestions. Below we provide responses to each weakness and question.
>
> **W1:** We would like to clarify the effectiveness of PLC from two aspects.
>
> **(1) The rationale behind Perturbation-Label Coupling.**
> The goal of PLC is to build strong associations between perturbations and labels, thus introducing shortcuts. To achieve this, we perform direct perturbation-label alignment by aligning the perturbation $\delta$, rather than the perturbed image $x+\delta$, with the label $y$, and we name this process as Perturbation-Label Coupling. Perturbations from the same class are clustering around the corresponding labels and are distant from other classes, tricking the model to classify based on the introduced shortcuts. Therefore, PLC improves UE's transferability by reducing reliance on specific data, playing a collaborative role in promoting unlearnability across various scenarios.
>
> **(2) The role of CLIP text encoder.**
>
> We note that the CLIP text encoder serves two roles and cannot be replaced with any random encoder.
> - **Anchors for perturbations to align with.** As mentioned above, the rationale behind PLC is to establish direct associations between perturbations and labels, where the plug-and-play CLIP text encoder holds exceptional zero-shot capability to encode class labels as anchors in the contrastive alignment paradigm, guiding the optimization of perturbations towards corresponding classes and away from other classes.
> - **Implicit connections with its vast image base.** Another reason for using CLIP text encoder derives from its superior cross-modal associations. It is known that CLIP's label encodings are closely associated with its extensive pre-trained image base. Therefore, by aligning perturbations with CLIP-encoded labels, the perturbation generator is able to implicitly connect with unseen images, which enhances its effectiveness across unseen classes.
>
>
> **W2:** We appreciate your observation. Our augmented images generated by the domain composer exhibit a similar degree of distributional shift and semantic consistency as the illustrative examples shown in Figure 1. Since we are not allowed to provide images in rebuttal, **we provide quantitative evaluations to confirm that the composer does introduce meaningful domain-level diversity without compromising class semantics**. We use PSNR (Pixel-level Fidelity) [1], SSIM (Structural Similarity) [2], and LPIPS (Perceptual Similarity) [3] to jointly assess the quality of diversified samples, ensuring they remain structurally consistent while exhibiting meaningful domain-level variation. We report average values over 100 randomly sampled augmented images, and include comparisons with both Gaussian-blurred images (as a weak baseline with few domain shifts) and the illustrative examples in Figure 1. Results are summarized below:
> |Metric|Generated Examples|GaussianBlur Examples|Examples in Figure 1|
> |:-:|:-:|:-:|:-:|
> |PSNR(dB)| 18.85| 23.20|16.35|
> |SSIM|0.6039|0.7290|0.5651|
> |LPIPS|0.5131|0.3379|0.5303|
>
> These results show that the augmentations generated by the domain composer match the examples in Figure 1 in terms of fidelity, distortion, and semantic similarity, which successfully diversifies training samples to optimize more transferable perturbations.
>
> **W3:** We value your advice and add evaluations against diffusion-based purification methods. As the code for BridgePure [4] is not publicly available, we adopt AVATAR [5], a recent representative diffusion-based defense, for comparison. As shown below, diffusion-based defense indeed exhibits strong capabilities in purifying perturbed images into trainable data, where the unlearnability of VTG degrades moderately. This is somewhat expected since VTG does not include specific mechanisms to resist diffusion-based defenses, a limitation shared by existing UE methods. Despite this challenge, VTG still **outperforms other UE methods** under AVATAR, demonstrating **superior robustness**. We consider the resistance of UEs to diffusion-based defense as an important future research direction.
>
> |Defense|Dataset|VTG|EMN|TAP|AR|REM|SHR|
> |:------:|:-----:|:------:|:------:|:------:|:------:|:------:|:------:|
> |AVATAR|CIFAR-10|**34.61**|90.95|90.71|91.57|88.49|85.69|
> ||CIFAR-100|**23.40**|65.73|64.99|64.54|64.88|58.52|
>
>
> **Q1:** We note that we use class-wise EMN perturbations and sample-wise TUE perturbations as baselines. The corresponding details for applying them to target datasets were mainly included in Supplementary 1.2. The implementation process can be categorized into two different scenarios.
>
> **(1) For target data with different numbers of classes or samples, we follow the interpolation operation introduced by TUE.**
>
> Specifically, if more classes are required, we interpolate two classes to create new class-wise perturbations:
>
> $\delta^* = \alpha \delta_i + (1 - \alpha) \delta_j, \text{where } y_i \neq y_j.$
>
> If more samples within one class are required, we interpolate two samples to create new sample-wise perturbations:
>
> $\delta^* = \alpha \delta_i + (1 - \alpha) \delta_j, \text{where } y_i = y_j.$
>
> The subscripts denote class indexes, and the number of newly created perturbations is controlled by varying $\alpha$, which is generally set as 0.5.
>
> **(2) For target data with different resolutions, we simply resample the perturbations to match the resolution of the target dataset.**
>
> [1] Scope of validity of PSNR in image/video quality assessment. Electronics Letters 2008.
>
> [2] Image quality assessment: from error visibility to structural similarity. IEEE Transactions on Image Processing 2004.
>
> [3] Investigating loss functions for extreme super-resolution. CVPRW 2020.
>
> [4] BridgePure: Limited Protection Leakage Can Break Black-Box Data Protection. arXiv 2024.
>
> [5] The devil’s advocate: Shattering the illusion of unexploitable data using diffusion models. IEEE SaTML 2024.

---

> > ### Comment · Reviewer_pQ4b · 2025-08-04
> >
> > I appreciate the authors’ response, which addressed my concerns. I will maintain my positive score. I look forward to seeing the corresponding updates in the final version.

---

> > > ### Author Response · Authors · 2025-08-04
> > >
> > > We sincerely thank Reviewer pQ4b for the recognition of our work and for the positive response. We are glad to see that your concerns have been addressed. The corresponding updates will be included in the revised version of the paper.

---

### Official Review · Reviewer_K9MH · 2025-07-03

**Clarity:** 3
**Significance:** 2
**Originality:** 3
**Rating:** 5
**Confidence:** 3

**Summary:**

This paper proposes a transferable generator to safeguard data across various conditions. The proposed method improves its generalizability to unseen scenarios by integrating adversarial domain augmentation into the generator’s training process. The generator becomes less dependent on data-specific representations, improving its transferability to unseen conditions. In addition, contrastive learning was leveraged to align perturbations with class labels. This approach reduces the generator’s reliance on data semantics, allowing the proposed method to produce unlearnable perturbations in a distribution-agnostic manner.

**Questions:**

please refer to the Weakness part

**Ethical Concerns:**

["NO or VERY MINOR ethics concerns only"]

**Final Justification:**

The rebuttal has addressed most of my concerns. After reviewing the other feedback, I am willing to raise my rating.

**Limitations:**

yes

**Quality:**

3

**Strengths And Weaknesses:**

Strength
1. The transferability of unlearnable examples is very important to real safeguard individual privacy. This paper introduces comprehensive transferable evaluation framework, including Intra-Domain, cross-domain, cross-task, and cross-space, and cross-architecture.
2. The proposed method uses the domain composer to generate novel domains lying outside the original distribution, so the generator becomes less dependent on data-specific representations, improving its transferability to unseen conditions.
3. The paper is well-written with a clear logical flow. The experimental results are promising, demonstrating the superior performance and wide applicability of the proposed approach across various scenarios.

Weakness
1. Although the quantitative evaluation results of the paper show a significant improvement in transferability of the proposed method, there is a lack of comparison in terms of visual quality of images or subjective evaluation metrics. This is crucial because, in addition to untrainable aspects, data quality is also very important in unlearnable examples.
2. The proposed method requires a generator to create perturbations at test time, resulting in computational overhead during deployment.
3. The proposed study includes empirical validation but lacks a thorough theoretical analysis of the diversity of generated samples or the influence of out-of-distribution samples on the transferability of the proposed method.

---

> ### Author Rebuttal · Authors · 2025-07-31
>
> We thank reviewer K9MH for the positive feedback and valuable suggestions. Below we provide responses to each weakness.
>
> **W1:** We emphasize that our perturbed images exhibit comparable visual quality to existing unlearnable example (UE) methods, and remain sufficiently clear for semantic interpretation. Due to rebuttal policy, we cannot present images to compare their visual quality. However, following your suggestion, we evaluate the quality of perturbed images from two aspects.
>
> **(1) Qualitative visualizations.** We have included some visualizations of the perturbed images in Figure 3 of Supplementary Section 1.4, where we compared the perturbed images with clean images across seven datasets, including CIFAR-10, CIFAR-100, MNIST, MNIST-M, SVHN, SYN, and PACS. The quality of perturbed images is generally acceptable, especially on CIFAR-10 and CIFAR-100. For comparison with other UE methods, please see visualizations in Figure 1 of [1].
> **(2) Quantitative Evaluations.** To further justify the visual quality of our method, we evaluate the structural similarity between perturbed and clean images using two widely adopted metrics: PSNR [2] and SSIM [3]. These metrics are known to reflect both low-level pixel fidelity and high-level structural consistency. As shown in the table below, we report the average values computed on 100 image samples of CIFAR10. The quality of our perturbed images is relatively comparable to existing UE methods.
>
> |Method | EMN   | LSP   | TUE   | GUE   | Ours  |
> | :---: |  :-:   | :-:   | :-:   | :-:   | :--:  |
> |PSNR (dB) ↑| 37.06 | 34.94 | 34.43 | **37.72** | 35.77 |
> | SSIM ↑  |  **0.9963** | 0.9930 | 0.9869| 0.9960 | 0.9894 |
>
> **W2:** We acknowledge that the generator structure introduces additional computational overhead during deployment, which we have identified as a limitation and discussed in the Conclusion section. In addition, we also compared the efficiency of VTG  against two other generator-based approaches, and the results are shown in Table 9 in our supplementary. It can be observed that **our generator is small in size (0.09M parameters)**. Owing to its lightweight structure, **it also generates perturbations efficiently (requiring only 0.4 ms per image)**. This indicates that VTG incurs minimal computational overhead at inference while effectively introducing unlearnability across diverse scenarios. We will include above inference cost analyses in the updated paper for better presentation.
>
> **W3:** We sincerely thank the reviewer for this insightful observation. We acknowledge that the theoretical analysis on the diversity of generated samples is indeed critical for a complete understanding of our transferability-focused UE framework. On the other hand, we would like to note that the theoretical analysis of UEs in the context of transferability is inherently challenging. To the best of our knowledge, no existing work has addressed this issue. We consider this a core focus of our future work.
>
> As a preliminary step toward this goal, we provide a theoretical understanding of our method through the lens of the Wasserstein distance between real samples and generated diversified samples.
>
> Specifically, given a predictor $f$, for any distributions $\mathbb{P}_a$ and $\mathbb{P}_b$, it can be shown that for any $0<\delta<1$, the following holds with probability at least $1-\delta$ [4]:
> $$\underset{(x,y)\sim\mathbb{P}_b}{\mathbb{E}}\mathcal{L}(f(x),y) \le \underset{(x,y)\sim\mathbb{\hat{P}}_a}{\mathbb{E}}\mathcal{L}(f(x),y)+W_d(\mathbb{\hat{P}}_a, \mathbb{\hat{P}}_b)+\mathcal{O}(\sqrt{\frac{\log (1/\delta)}{N_a}},\sqrt{\frac{\log (1/\delta)}{N_b}}),$$
> where $\mathcal{L}$ is a loss function (e.g., cross-entropy loss), $\mathbb{\hat{P}}$ is the empirical counterpart of $\mathbb{{P}}$ (i.e., a data set sampled from $\mathbb{{P}}$), $N_a$ and $N_b$ are, respectively, the sample size of $\mathbb{\hat{P}}_a$ and $\mathbb{\hat{P}}_b$. The inequality reveals that we can upper bound the expected loss of $\mathbb{{P}}_b$ in terms of the empirical loss on $\mathbb{\hat{P}}_a$ and their empirical Wasserstein distance $W_d(\mathbb{\hat{P}}_a, \mathbb{\hat{P}}_b)$.
>
> For the problem studied in this work, however, we cannot estimate $W_d(\mathbb{\hat{P}}_a, \mathbb{\hat{P}}_b)$, since we only have a training set $\mathbb{\hat{P}}_a$, but the test data $\mathbb{\hat{P}}_b$ is unknown. As our objective is to enhance the transferability in various distinct scenarios, we address this issue by training a domain composer $\mathcal{C}\_{\mu}$ to synthesize the worst-case out-of-distribution [5, 6] samples:
> $${\mu} = \underset{\mu}{\arg\max} \mathcal{W}_d(\mathcal{C}\_{\mu}({x}), {x}).$$ In practice, the computation of $\mathcal{W}_d(\mathcal{C}\_{\mu}({x}),x)$ involves a linear programming problem whose cost is prohibitive, and therefore we adopt the Sinkhorn distance [7, 8] as defined in Eq. (3) in our paper. Additionally, it can be shown that under certain conditions, the Sinkhorn distance is equivalent to the Wasserstein distance [7, 8]. Putting all together, we can show that, the following holds with probability at least $1-\delta$:
> $$\underset{x\sim \mathcal{C}\_\mu(x)}{\mathbb{E}}\mathcal{L}(f(x),y) \le \underset{(x,y)\sim\mathbb{\hat{P}}_a}{\mathbb{E}}\mathcal{L}(f(x),y)+W_d(\mathcal{C}\_\mu(x), x)+\mathcal{O}(\sqrt{\frac{\log (1/\delta)}{N_a}}).        \qquad     (*)$$
>
> In summary, our domain composer leverages Wasserstein distance to generate worst-case samples and its diversity can be roughly analyzed via Eq. (*) by examining the loss on synthetic samples. On the other hand, we note that this analysis is only an initial step. For example, it does not examine how the composer $\mathcal{C}_\mu$ affects the transferability of the noise $\delta$. Besides, conducting a lower-bound analysis of the loss is also a promising direction for improving our understanding of sample diversity and noise transferability.
>
>
> [1] Detecting and Corrupting Convolution-based Unlearnable Examples. AAAI 2025.
>
> [2] Scope of validity of PSNR in image/video quality assessment. Electronics Letters 2008.
>
> [3] Image quality assessment: from error visibility to structural similarity. IEEE Transactions on Image Processing 2004.
>
> [4] Theoretical Analysis of Domain Adaptation with Optimal Transport. ECML 2017
>
> [5] Certifying Some Distributional Robustness with Principled Adversarial Training. ICLR 2018
>
> [6] Out-of-Domain Generalization From a Single Source: An Uncertainty Quantification Approach. IEEE TPAMI 2022.
>
> [7] Sinkhorn Distances: Lightspeed Computation of Optimal Transport. NeurIPS 2013.
>
> [8] Learning Generative Models with Sinkhorn Divergences. AISTATS 2018.

---

### Note · Authors · 2025-08-13

Dear Reviewers/AC/SAC/PC,

We sincerely thank the Reviewers and the AC for their effort and insightful feedback. Below, we summarize the contributions recognized by the reviewers and our responses to the main concerns.

## **Summary of Contributions**

We are encouraged that the reviewers recognized the key strengths of our work, including conducting **the first comprehensive study** on the transferability of UEs **[K9MH, pQ4b, Dthy, bdeR]**, and introducing a **novel domain composer** with Adversarial Domain Augmentation to improve generalization to unseen scenarios **[K9MH, pQ4b, mhbh]**. Reviewers also found our experimental results to be **promising and impressive** **[K9MH, pQ4b, Dthy]**, our **ablation studies** to be useful **[Dthy]**, our **resilience** against common defense strategies to be **strong** **[bdeR]**, and our paper to be **well-written and well-organized** **[K9MH, pQ4b, Dthy, bdeR]**.

## **How we addressed the primary concerns**

1. Included new **data quality evaluations** for perturbed images and augmented images **[K9MH, pQ4b, Dthy]**.
2. Provided a **preliminary theoretical understanding** through the lens of the Wasserstein distance between real samples and generated diversified samples **[K9MH]**.
3. Added evaluations against **diffusion-based defenses**, ablation study on **generator depth**, different **augmentation strategies**, and different **vision-language encoders** **[pQ4b, Dthy, bdeR]**.
4. Added empirical analyses on **why the generated perturbations work** **[bdeR]**.
5. **Clarified key concerns raised by reviewers**:
    - Provided that **visualizations** and **inference efficiency** are already included in Supplementary **[K9MH, Dthy]**.
    - Explained the **motivation** behind each module and how they work **jointly** to achieve the overall objective **[pQ4b, Dthy, bdeR]**.
    - Clarified the **comprehensive comparison** with both class-wise and sample-wise baselines, and added two explicitly class-wise baselines **[bdeR]**.
    - Clarified the **distinction** between UEs and data poisoning attacks, and showed that VTG **does not increase privacy risk**, as supported by MIA results **[mhbh]**.

We view these changes as constructive refinements that enhance the clarity, coherence, and overall quality of our work, without altering our core contributions. We have carefully addressed all concerns raised by reviewers and will incorporate all updates into the updated paper.

Best,

Submission 26468 Authors

---

### Decision · Program_Chairs · 2025-09-17

**Decision:**

Accept (poster)

**Comment:**

The paper discusses “unlearnable examples” that continue to work outside their training setup, i.e., across new datasets, tasks, resolutions, and architectures. This is the primary contribution of the paper: taking the concept of unlearnable examples and making it work across diverse settings. It trains a Versatile Transferable Generator (VTG) with two key ideas: adversarial domain augmentation to expose the generator to OOD shifts, and perturbation–label coupling (via contrastive alignment) so models latch onto the perturbation rather than semantics, making the result largely distribution-agnostic. In rigorous transfer tests spanning five settings, VTG consistently outperforms prior UE methods.

Most reviewers agree that the paper shows strong transferability evaluations (intra-domain, cross-domain/task/space/architecture) with consistently strong numbers. The method components are well-motivated and empirically supported. The writing is also generally clear and well organised.

On the weakness front, multiple reviewers asked for a theoretical explanation of why the method works. The authors attempted to provide a Wasserstein-distance-based explanation, upon which the reviewer increased their score. However, after reading the authors’ response, I think the provided theoretical expressions do not shed much light on the success of the method. The inequalities stated in the rebuttal hold under several non-trivial assumptions on the data, the distribution shift, the model, and the loss function. The equivalence between OT and Sinkhorn is also not as clear as the authors stated. Nevertheless, most reviewers agreed that the absence of theory is not a concern here and I agree with that. Nevertheless, I would suggest that in a future draft if these mathematical arguments were included, they be made more rigorous.

A second weakness discussed by multiple reviewers was using test accuracy as the only metric. The authors rebutted by saying this is standard in the UE literature. However, there are two aspects to this criticism, and I agree with both of them:

1. If only test accuracy were used without any constraint on the perturbation magnitude, this would make the problem trivial. The authors rebut by saying the perturbations are indeed small, but I think this should be a standard part of the evaluations, not something shown selectively in response to reviews.

2. More importantly, the authors say that the main difference between this and the classical poisoning literature (response to reviewer mhbh) is that the goal of unlearnable examples is to protect privacy. This appears to be the biggest flaw to me because (a) the paper does not mention what privacy means in this context, (b) the measured quantities (e.g., test accuracy) are not valid indicators of privacy: it is very easy to have bad accuracy on a set but still leak information about that set, and (c) in fact, as reviewer mhbh highlights, adding noisy ``shortcuts'' to images so that the model remembers those shortcuts is a clear violation of privacy. The model will remember the shortcuts and hence the associated image.

Nevertheless, despite the above two weaknesses, I believe the contributions of the paper outweigh these weaknesses. The main contribution, namely generalising the concept of "unlearnable examples" to diverse settings and providing a strong and comprehensive experimental evaluation to make that point was received very well by all reviewers. Additionally, the proposed algorithm is intuitive and cleverly combines techniques present in the ood robustness literature. Given this, I recommend acceptance.